# Simulating second-generation herbaceous bioenergy crop yield using the global hydrological model H08 (v.bio1)

Zhipin Ai[1], Naota Hanasaki[1], Vera Heck[2], Tomoko Hasegawa[3], Shinichiro Fujimori[4]

[1]Center for Climate Change Adaptation, National Institute for Environmental Studies, 16-2, Onogawa, Tsukuba 305-8506, Japan

[2]Potsdam Institute for Climate Impact Research, Telegraphenberg A 31, Potsdam 14473, Germany

[3]Department of Civil and Environmental Engineering, Ritsumeikan University, 56-1, Toji-in Kitamachi, Kita-ku, Kyoto 603-8577, Japan

[4]Department of Environmental Engineering, Kyoto University, Building C1-3, C-cluster, Kyoto-Daigaku-Katsura, Nishikyo-ku, Kyoto 615-8504, Japan

*Correspondence to*: Zhipin Ai (ai.zhipin@nies.go.jp)

**Abstract.** Large-scale deployment of bioenergy plantations would have adverse effects on water resources. There is an increasing need to ensure the appropriate inclusion of the bioenergy crops in global hydrological models. Here, through parameter calibration and algorithm improvement, we enhanced the global hydrological model H08 to simulate the bioenergy yield from two dedicated herbaceous bioenergy crops, *Miscanthus* and switchgrass. Site-specific evaluations showed that the enhanced model had the ability to simulate yield for both *Miscanthus* and switchgrass, with the calibrated yields being well within the ranges of the observed yield. Independent country-specific evaluations further confirmed the performance of the H08 (v.bio1). Using this improved model, we found that unconstrained irrigation more than doubled the yield under rainfed condition, but reduced the water use efficiency (WUE) by 32% globally. With irrigation, the yield in dry climate zones can exceed the rainfed yields in tropical climate zones. Nevertheless, due to the low water consumption in tropical areas, the highest WUE was found in tropical climate zones, regardless of whether the crop was irrigated. Our enhanced model provides a new tool for the future assessment of bioenergy–water tradeoffs.

## 1 Introduction

Bioenergy with carbon capture and storage technology enables the production of energy without carbon emissions, while sequestering carbon dioxide from the atmosphere, producing negative emissions. Therefore, bioenergy is considered an important technology in the push to achieve the 2-degree climate target (Smith et al., 2015). With ambitious climate policies, the demand for bioenergy in 2100 could reach 200–400 EJ per year, based on recent predictions (Rose et al., 2013; Bauer et al., 2018). However, large-scale plantating of bioenergy crops requires water consumption to be doubled or even tripled, which would exacerbate the future water scarcity (Beringer et al., 2011; Bonsch et al., 2016; Hejazi et al., 2015; Yamagata et al., 2018). Therefore, representation of bioenergy crops in global hydrological models is critical in elucidating the possible side effects of large-scale implementation of bioenergy.

Second-generation bioenergy crops, such as *Miscanthus* and switchgrass, are generally regarded as a dedicated bioenergy source due to their high yield potential and lack of direct competition with food production (Beringer et al., 2011; Yamagata et al., 2018; Wu et al., 2019). This is because *Miscanthus* and switchgrass are rhizomatous perennial C4 grasses, which have a high photosynthesis efficiency (Trybula et al., 2015). These two crops have been included in a series of models including Lund–Potsdam–Jena managed Land (LPJml) (Beringer et al., 2011; Bondeau et al., 2007), H08 (Yamagata et al., 2018), ORCHIDEE (Li et al., 2018), the High-Performance Computing Environmental Policy Integrated Climate model (HPC-EPIC) (Kang et al., 2014; Nichols et al., 2011), the Community Land Model (version 5) (CLM5) (Cheng et al., 2020), MISCANMOD (Clifton-Brown et al., 2000; 2004), MISCANFOR (Hastings et al., 2009), Agricultural Production Systems Simulator (APSIM) (Ojeda et al., 2017), and the Soil & Water Assessment Tool (SWAT) (Trybula et al., 2015). However, among these models, only a few, such as LPJml, H08, and CLM5 include the global implementation of schemes for irrigation, river routing or water withdrawal. This severely limits the application of the models to address the global bioenergy−water tradeoffs or synergies.

To the best of our knowledge, LPJml was the first global model that includes both bioenergy and the water cycle. It has therefore been widely used to quantify the effects on water of large-scale plantating of bioenergy crops in many previous studies (Beringer et al., 2011; Heck et al., 2016; 2018; Bonsch et al., 2016; Janes et al., 2018; Stenzel et al., 2019). However, it should be noted that *Miscanthus* and switchgrass are not distinguished in LPLml, which instead uses a C4 grass to parameterize them. A separate parametrization for the two bioenergy crops could enhance the bioenergy simulation since they showed totally different plant characteristics and crop yield (Heaton et al., 2008; Trybula et al., 2015; Li et al., 2018). CLM5 has been improved and validated for simulating *Miscanthus* and switchgrass separately based on observations at the University of Illinois Energy Farm (Cheng et al., 2020), but a global validation or application has not been reported. H08 is a global hydrological model that considers human activities, including reservoir operation, aqueduct water transfer, seawater desalination, and water abstraction for irrigation, industry, and municipal use (Hanasaki et al., 2008a, 2008b, 2010a, 2018a, 2018b). The first use of H08 to simulate the bioenergy crop yield was reported in an impact assessment of the effects of BECCS on water, land, and ecosystem services (Yamagata et al., 2018). Using an identical model to that of Yamagata et al. (2018), another recent study also used H08 estimates of *Miscanthus* and switchgrass yield to predict global advanced bioenergy potential (Wu et al., 2019). Based on the work of Yamagata et al. (2018), here we improved the bioenergy crop simulation in H08 by performing a systematic parameter calibration for both *Miscanthus* and switchgrass using the best available data.

The objective of this study was to enhance and validate the ability of H08 to simulate the second-generation herbaceous bioenergy crop yield. The following sections of this paper will: 1) describe the default biophysical process of the crop module in H08, 2) explain the enhancement of H08 for *Miscanthus* and switchgrass, 3) evaluate the enhanced performance of the model in simulating yields for *Miscanthus* and switchgrass, 4) map the spatial distributions of the yield of *Miscanthus* and

switchgrass, and 5) illustrate the effects of irrigation on the yield, water consumption, and WUE (defined here as the ratio of yield to water consumption) of *Miscanthus* and switchgrass.

## 2 Materials and methods

### 2.1 H08 and its crop module

H08 is a global hydrological model that can simulate the basic natural and anthropogenic hydrological processes as well as crop growth at a spatial resolution of 0.5° and at a daily interval (Hanasaki et al., 2008a, 2008b). Main variables related to the water cycle, such as river discharge, terrestrial water storage, and water withdrawal have been thoroughly validated in a series of previous studies (Hanasaki et al., 2008a, 2008b, 2018). H08 is consist of six submodules. The six submodules (land surface hydrology, river routing, crop growth, reservoir operation, environmental flow requirements, and anthropogenic water withdrawal) are coupled in a unique way (Fig. 1a). The land surface module can simulate the main water cycle components, such as evapotranspiration and runoff. The former is used in the crop module, and the latter is used in the river routing and environmental flow modules. The agricultural water demand simulated by the crop module and the streamflow simulated by the river routing and reservoir operation modules finally enter into the withdrawal module. Note that the crop module is independent, except for the water stress calculations, which require evapotranspiration and potential evapotranspiration inputs from the land surface hydrology module.

Figure 1b shows the basic biophysical process of the crop module in H08. The biomass accumulation is based on Monteith et al. (1977). The crop phenology development is based on daily heat unit accumulation theory. The harvest index is used to partition the grain yield. Regulating factors, including water and air temperature, are used to constrain the yield variation. The crop module can simulate the potential yield, crop calendar, and irrigation water consumption for 18 crops, including barley, cassava, cotton, peanut, maize, millet, oil palm, potato, pulses, rape, rice, rye, sorghum, soybean, sugar beet, sugarcane, sunflower, and wheat. The parameters for these crops were taken from those of the SWAT model. To better reflect the agronomy practice, H08 divides each simulation cell into four sub-cells: rainfed, single-irrigated, double-irrigated, and other (i.e., non-agricultural land uses). Irrigation in H08 is defined as the supply of water other than precipitation to maintain soil moisture above 75% of field capacity during the cropping period. To clarify this as regards the function of the parameters we calibrated below, here we describe the algorithms in the crop module of H08. The crop module of H08 accumulates daily heat units ($Huna(t)$), which are expressed as the daily mean air temperature ($T_a$) greater than the plant's specific base temperature ($Tb$; given as a crop-specific parameter):

$$Huna(t) = T_a - Tb \tag{1}$$

Then the heat unit index ($Ihun$) is calculated as the ratio of accumulated daily heat units $\sum Huna(t)$ and the potential heat unit ($Hun$):

$$Ihun = \frac{\sum Huna(t)}{Hun} \tag{2}$$

When the accumulated daily heat units $\sum Huna(t)$ reach the potential heat unit ($Hun$) required for the maturity of the crop, the crop is mature and is harvested. During the growth period, the daily increase in biomass ($\Delta B$) is calculated using a simple photosynthesis model:

$$\Delta B = be * PAR * REGF \tag{3}$$

where $be$ is radiation use efficiency, $PAR$ is photosynthetically active radiation, and $REGF$ is the crop regulating factor. $PAR$ is calculated using shortwave radiation ($Rs$) and leaf area index ($LAI$) as follows:

$$PAR = 0.02092 * Rs * [1 - \exp(-0.65 * LAI)] \tag{4}$$

LAI is calculated according to the growth stage indicated by $Ihun$, if $Ihun < \lfloor dpl1 \rfloor * 0.01$,

$$LAI = \frac{(dpl1 - \lfloor dpl1 \rfloor) * Ihun}{\lfloor dpl1 \rfloor * 0.01} * blai \tag{5}$$

if $\lfloor dpl1 \rfloor * 0.01 \leq Ihun < \lfloor dpl2 \rfloor * 0.01$,

$$LAI = \left\{ (dpl1 - \lfloor dpl1 \rfloor) + \frac{[(dpl2 - \lfloor dpl2 \rfloor) - (dpl1 - \lfloor dpl1 \rfloor)] * (Ihun - \lfloor dpl1 \rfloor * 0.01)}{\lfloor dpl2 \rfloor * 0.01 - \lfloor dpl1 \rfloor * 0.01} \right\} * blai \tag{6}$$

if $\lfloor dpl2 \rfloor * 0.01 \leq Ihun < dlai$,

$$LAI = \left\{ (dpl2 - \lfloor dpl2 \rfloor) + \frac{[1 - (dpl2 - \lfloor dpl2 \rfloor)] * (Ihun - \lfloor dpl2 \rfloor * 0.01)}{dlai - \lfloor dpl2 \rfloor * 0.01} \right\} * blai \tag{7}$$

if $dlai < Ihun$,

$$LAI = 16 * blai (1 - Ihun)^2 \tag{8}$$

where dpl1 and dpl2 are two complex numbers (see the definition in Table 1), blai is the maximum leaf area index.

$REGF$ is calculated as:

$$REGF = \min(Ts, Ws, Ns, Ps) \tag{9}$$

where $Ts, Ws, Ns,$ and $Ps$ are the respective stress factors for temperature, water, nitrogen, and phosphorous. Temperature stress ($Ts$) is calculated as an asymmetrical function according to the relationship between air temperature ($T_a$) and optimal temperature ($To$). When air temperature is below (or equal) the optimal temperature ($To$), $Ts$ is calculated as:

$$Ts = exp\{ln(0.9) * [\frac{Ctsl(To - T_a)}{T_a}]^2\} \tag{10}$$

where $Ctsl$ is the temperature stress parameter for temperature below $To$, and is calculated as:

$$Ctsl = \frac{To + Tb}{To - Tb} \tag{11}$$

When air temperature is above the optimal temperature, $Ts$ is calculated as:

$$Ts = exp\{ln(0.9) * [\frac{(To - T_a)}{Ctsh}]^2\} \tag{12}$$

where $Ctsh$ is the temperature stress parameter for temperature below $To$, and is calculated as:

$$Ctsh = 2 * To - T_a - Tb \tag{13}$$

Water stress ($Ws$) is calculated as the ratio of actual evapotranspiration ($Ea$) to potential evapotranspiration ($Ep$) as:

$$Ws = \frac{Ea}{Ep} \tag{14}$$

The crop yield ($Yld$) is finally estimated from the aboveground biomass ($Bag$) using the crop-specific harvest index ($Harvest$) at the harvesting date as:

$$Bag = [1 - (0.4 - 0.2 * Ihun)] \sum \Delta B \tag{15}$$

$$Yld = Harvest * \frac{WSF}{WSF + \exp(6.117 - 0.086 * WSF)} * Bag \tag{16}$$

where $WSF$ is the ratio of $SWU$ (the accumulated actual plant evapotranspiration in the second half of the growing season), and $SWP$ (the accumulated potential evapotranspiration in the second half of the growing season):

$$WSF = \frac{SWU}{SWP} * 100 \tag{17}$$

**2.2 Enhancement of H08 for *Miscanthus* and switchgrass**

The original bioenergy crop implementation in H08 (Yamagata et al., 2018) was conducted in two steps. First, crop parameters (see the old values in Table 2) for *Miscanthus* (refer to *Miscanthus giganteus* in this study) and switchgrass (refer to *Panicum virgatum* in this study) were adopted based on the settings from the SWAT model 2012 version (Arnold et al., 2013). However, the default parameters did not reflect the characteristics for *Miscanthus* and switchgrass well, which could lead to serious bias based on the result in Trybula et al. (2015). Second, maturity was defined by either undergoing an autumn freeze (i.e., the air temperature was below the minimum temperature for growth) or the exceedance of the maximum of 300 continuous days of growth. Because both *Miscanthus* and switchgrass are perennial, the potential heat unit was set as unlimited (see the old values in Table 2). However, this unlimited potential heat unit is far from the observations (see the new values in Table 2) reported by Trybula et al. (2015). Here, further enhancements were made as follows. First, we changed the leaf area development curve by adopting the potential heat unit (Hun) and leaf area related parameters (dpl1 and dpl2) proposed by Trybula et al. (2015). The potential heat unit can determine both the total cropping days and the leaf development. Here, we set the values at 1,830 and 1,400 degrees for *Miscanthus* and switchgrass, respectively, as recommended by Trybula et al. (2015) based on their field observations. The dpl1 and dpl2 parameters (see Table 1), which were used for determining the leaf development curve, were also changed to the values suggested by Trybula et al. (2015). This modification substantially changed the original heat unit index (Ihun) and the development of the leaf area index curve. Second, we modified the algorithm for water stress that was used to regulate the radiation use efficiency. We took the ratio of actual evapotranspiration to potential evapotranspiration as the water stress factor for any point in the simulation, similar to the description of the soil moisture deficit used in other studies (Anderson et al., 2007; Yao et al., 2010). Third, we added a new output variable for the water consumption of *Miscanthus* and switchgrass to analyze the water consumption and WUE in the crop sub-module. Fourth, we introduced the Köppen climate classification (see Fig. 2) into the source code to provide possible climate-specific analyses. Finally, we conducted parameter calibrations with the best available data. The calibration process is presented below, and the finalized parameter settings are given in Table 2.

We conducted a calibration with five important parameters, the radiation use efficiency (be), maximum leaf area index (blai), base temperature (Tb), maximum daily accumulation of temperature (Hunmax), and minimum temperature for planting (TSAW). The specific parameter ranges and steps set in the calibration process are shown in Table 3. In total, 1,944 simulations

were conducted for *Miscanthus* and switchgrass to test all combinations of the parameter sets. The simulations were conducted with the averaged daily meteorology data from WFDEI (1979-2016) for two reasons. First, using multi-year averaged metrology input can exclude the effect of extreme climate (low temperatures in early spring and late autumn) on the yield and this is recommended in the H08 manual (Hanasaki et al., 2010). Second, it can largely save the computation storage. The best parameter sets were selected using two steps: first, the lowest root mean square error (RMSE), and second, the highest correlation coefficient (R) of the simulated and observed yields within the lowest RMSE domain. Additional information on how these parameters affect the model can be found in the equations described in Section 2.1.

## 2.3 Model input data

The WATCH-Forcing-Data-ERA-Interim (WFDEI) global meteorological data (Weedon et al., 2014) from 1979 to 2016 were used in all simulations. The WFDEI data were based on the methodology used for WATer and global CHange (WATCH) forcing data by utilizing ERA-Interim global reanalysis data. The data cover the whole globe at a spatial resolution of 0.5°. Eight daily meteorological variables (air temperature, wind speed, air pressure, specific humidity, rainfall, snowfall, and downward shortwave and longwave radiation) were used to run H08. Another meteorological dataset for the period 1979–2013 in S14FD (Iizumi et al. 2017) with the same spatial resolution was also used to check the stability of results to input meteorological data.

## 2.4 Yield data

To independently calibrate and validate the performance of H08 in simulating the bioenergy yield, we collected and compiled up-to-date site-specific (varied from 1986 to 2011) and country-specific (varied from 1960 to 2010) yield data from both observations and simulations (Clifton-Brown et al., 2004; Searle and Malins, 2014; Heck et al., 2016; Kang et al., 2014; Li et al., 2018a). For *Miscanthus*, the yield data used covered 72 sites (64 rainfed and 8 irrigated; observed) and 15 countries (simulated). The simulated country-specific data is from MISCANMOD and LPJml. For switchgrass, the yield data used covered 57 sites (55 rainfed and 2 irrigated; observed) and 16 countries (simulated). The simulated country-specific data are from HPC-EPIC and LPJml. A map showing the locations of the majority of sites under the rainfed condition and the corresponding climate zone is presented in Fig. 2. The data sites were predominantly distributed in Europe and the US. It should be noted that the sites are generally located in temperate and continental climate zones, with few located in the tropics and dry climate zones. Detailed lists of the sites from which the yields of *Miscanthus* and switchgrass were reported are documented in Tables S1 and S2 (for the rainfed condition) and Table S3 (for the irrigated condition) in the supplementary material.

A global yield map of *Miscanthus* and switchgrass that was generated using a random-forest algorithm (Li et al., 2020) was also used to compare the results. This yield map provides a benchmark for evaluating model performance because it is largely constrained by the observed yield ranges, denoting the yields achievable under current technologies (Li et al., 2020).

## 2.5 Simulation setting

After calibration, four different kinds of simulation were run with different purposes. The first simulation was conducted using the original model without irrigation to investigate its performance. The second simulation was conducted using the enhanced model without irrigation to investigate its performance under rainfed condition. The third simulation was conducted using the enhanced model with irrigation to investigate its performance under irrigated condition. These three simulations were conducted at a daily scale with annual meteorological data from WFDEI for the period 1979–2016. The last simulation was conducted using identical model settings to the third one, except using different meteorological data from S14FD for the period 1979–2013. Note that irrigation in this study means uniform unconstrained irrigation.

## 2.6 Water use efficiency

Water use efficiency (WUE) is an important indicator that shows the efficiency of crops in using water to produce biomass (Ai et al., 2020), which is useful in evaluating bioenergy crop performance (Zeri et al., 2013). Here, WUE is calculated as the ratio of yield to water consumption:

$$WUE = \frac{yield}{water\ consumption} \tag{18}$$

where yield and water consumption refer to the bioenergy crop yield (kg ha$^{-1}$ yr$^{-1}$) and the corresponding water consumption (mm yr$^{-1}$) of *Miscanthus* and switchgrass.

## 2.7 Sensitivity analysis

To see the sensitivity of the calibrated variables to the yield simulation, we calculated the sensitivity index (S) (Cheng et al., 2020) value for each variable:

$$S = \sum \overline{\left| \frac{(V_s - V_{ref})/V_{ref}}{(P_s - P_{ref})/P_{ref}} \right|} \tag{19}$$

where $V_s$ and $V_{ref}$ is the calculated RMSE of the simulated and observed yields for the corresponding calibration simulations and the finalized simulation (with final fixed parameters in Table 2), $P_s$ and $p_{ref}$ are the parameter values for the corresponding calibration simulations and the finalized simulation.

## 3 Results and discussion

### 3.1 Parameter calibration

The variation in RMSE and R for all 1944 simulations is presented in Fig. S1. Both RMSE and R have large ranges. Based on the optimal values of RMSE (4.68 and 3.16 Mg ha$^{-1}$ yr$^{-1}$ for *Miscanthus* and switchgrass, respectively) and R (0.67 and 0.53 for *Miscanthus* and switchgrass, respectively), we finalized the parameter set as shown in Table 2. The simulations presented in the table are for rainfed conditions because only afew sites were irrigated. The radiation use efficiency values were set at 38 and 22 (g MJ$^{-1}$ × 10) for *Miscanthus* and switchgrass, respectively. These values are similar to those of Trybula et al. (2015), who recommended values of 41 (g MJ$^{-1}$ × 10) for *Miscanthus* and 17 (g MJ$^{-1}$ × 10) for switchgrass. The base temperatures were calibrated to be 8 and 10°C for *Miscanthus* and switchgrass, respectively. The base temperature is sensitive to the crop growing days. Ranges from 7 to 10°C for *Miscanthus* and from 8 to 12°C for upland switchgrass were suggested by Trybula et al. (2015). The calibrated values are within the above ranges. The maximum leaf area indices were calibrated at 11 and 8 for *Miscanthus* and switchgrass, respectively; these values were identical to those suggested by Trybula et al. (2015). Of the five parameters we calibrated, radiation use efficiency was the most sensitive parameter to the result, followed by the base temperature (see Table S5), this is consistent to the result of Trybula et al. (2015). As shown in Fig.3, the calibrated parameters performed well, since the scatter points are well distributed along the 1:1 line.

### 3.2 Site-specific performance of enhanced H08

An overview of the performance of the enhanced H08 is provided in Fig. 4. The simulated yield is the annual average from 1986 to 2011. Points in a scatterplot comparing simulated yields derived from the enhanced H08 with observed yields are well distributed along the 1:1 line. It can be seen that the performance of the enhanced H08 was improved over that of the original H08. For *Miscanthus*, the bias of original model ranged from –84% to 80% with a mean of –52%, while the bias of the enhanced model ranged from –59% to 53% with a mean of –9%. For switchgrass, the bias for original model ranged from –78% to 338% with a mean of 25%, while the bias for the enhanced model ranged from –52% to 109% with a mean of –7%. Note that it also shows a tendency toward underestimation for some sites, especially for *Miscanthus*. More detailed site-specific results are shown in Figs. 5a (*Miscanthus*) and Fig. 5b (switchgrass). To depict the uncertainties in the observed yield, the

minimum and maximum observed yields are shown as error bars in Fig. 5. It was found that the simulated yields were within or close to the range of the observed yields. The simulated relative error was randomly distributed, was substantially smaller than the range of the observed yields, and showed no climatic bias. This implies that the combination of the Hun identified by Tryubula et al (2015) and the calibrated parameters of this study are valid for climate zones other than that of the midwestern US, where the Hun was observed. We also investigated the performance under the irrigated condition (shown in Fig. 6). We used the reported observed yields for ten sites globally (Table S3). We found that the simulated yields were within or close to the observed yields for five sites located in China, the UK, and France (see Table S3), but were overestimated for the remaining sites. This was due to the assumption of irrigation. H08 assumes that irrigation is fully applied to crops and hence the yield represents the maximum potential yield under irrigation condition. Therefore, if the reported yield is within the range of the simulated yield between rainfed and irrigated conditions, it is considered reasonable. This was found to be the case, as shown in Fig. 6. To investigate the uncertainty in the meteorological data, a simulation using other meteorological data from the S14FD dataset (Iizumi et al. 2017) was conducted; the results are compared in Fig. S2. The comparison showed that the WFDEI driven result was very similar to that obtained with the S14FD data.

**3.3 Country-specific performance of enhanced H08**

Figure 7 compares the yield simulated by the enhanced H08 with the collected independent country-specific yields simulated by MISCANMOD (Clifton-Brown et al., 2004), HPC-EPIC (Kang et al., 2014), and LPJmL (Heck et al., 2016). Here, the yield was simulated under rainfed conditions. The periods of climate data used as inputs were 1960–1990, 1980–2010, and 1982–2005 for MISCANMOD, HPC-EPIC, and LPJmL, respectively. Here, the comparisons were conducted using exactly the same period as that of HPC-EPIC and LPJmL. For MISCANMOD, however, we used the data from 1979–1990 due to data availability. For *Miscanthus*, the correlation coefficient of the yield simulated by H08 and MISCANMOD in the scatterplot (Fig. 7d) was 0.40. A t-test showed that the correlation was not significant at the 0.01 level. For consistency with the yield collected by MISCANMOD, any area within a country where the yield was less than 10 Mg ha$^{-1}$ yr$^{-1}$ was excluded from the analyses. Also, the land available for calculations was set as 10% of the pastureland and cropland. For switchgrass, the correlation coefficient of the yield simulated by H08 and HPC-EPIC in the scatterplot (Fig. 7e) was 0.80. A t-test showed that the correlation was significant at the 0.01 level. This indicates that the spatial pattern of the yield simulated by H08 was similar to that of HPC-EPIC. For example, both models produced high yields in Brazil, Colombia, Mozambique, and Madagascar, and low yields were found in Australia and Mongolia.

*Miscanthus* and switchgrass are not distinguished in LPJmL, and we therefore compared the mixed (mean, *Miscanthus* and switchgrass) yield of *Miscanthus* and switchgrass simulated by H08 and the C$_4$ grass yield simulated by LPJmL. The correlation coefficient of the yield simulated by H08 and LPJmL in the scatterplot (Fig. 7f) was 0.77. A t-test showed that the correlation was significant at the 0.01 level. An additional comparison under the irrigated condition is presented in Fig. S3. The correlation coefficient of the yield simulated by H08 and LPJmL, as shown in the scatterplot (Fig. S3), was 0.95. A t-test showed that the correlation was significant at the 0.01 level. The difference was mainly due to Colombia, Sudan, Mozambique, and Mexico, which are located in tropical zones. The difference in these countries was generally equal to the range of H08. For example, as shown in Fig. 7c, the yield in Colombia simulated by LPJmL was equal to the *Miscanthus* yield simulated by H08 (upper error bar). A separate comparison of the ensemble yield simulated by LPJmL, and the yield of *Miscanthus* and switchgrass simulated by H08 under both rainfed and irrigated conditions, is presented in Fig. S4. It can be seen that the yield of *Miscanthus* simulated by H08 was closer to the yield simulated by LPJml, which indicates that the LPJml-simulated yield was more likely to represent *Miscanthus*. This can also be inferred from the validation results (Fig. 1a) in Heck et al. (2016) since the LPJml-simulated yield is close to the yield of *Miscanthus* compared to those of switchgrass. It was difficult to

determine which model performed better due to the lack of observed data in tropical zones. This also indirectly indicated the relatively large uncertainty of the existing simulations in tropical zones (Kang et al., 2014).

The differences in model structure, use of specific algorithms, and the input climate data (different periods and sources) can induce differences in the yield simulated by MISCANMOD, HPC-EPIC, LPJmL, and H08. With regard to model structure, MISCANMOD uses a Kriging interpolation method to derive the spatial yield from the original site yield, whereas H08, LPJmL, and HPC-EPIC use grid-based calculations. H08 considers the single harvest system in tropical areas, whereas LPJml

considers a multiple harvest system. With regard to the specific algorithms used, the water stress used to regulate radiation-use efficiency varies considerably among the models. Note that the differences in meteorological data sources and spatial-temporal resolution would also contribute to these differences.

### 3.4 Further evaluation of the performance of enhanced H08

As shown in Fig. 8, we compared our simulation with the latest available global bioenergy crop yield map, generated from observations using a random-forest (RF) algorithm (Li et al., 2020). This RF yield map provides a benchmark for evaluating model performance because it is largely constrained by the observed yield ranges, denoting the yields achievable under current technologies (Li et al., 2020). As shown in Fig. 8a and Fig. 8b, there were small differences between our estimated yield and RF yield for switchgrass, whereas larger differences were found for *Miscanthus*, especially in tropical regions. There is a

similar case for ORCHIDEE, as shown in Fig. S21 in Li et al. (2020). We also compared the differences in the mean values for *Miscanthus* and switchgrass because they are not distinguished in LPJmL. As shown in Fig. 8c and Fig. 8d, the differences between our estimations and the RF yields were generally lower than those between the LPJml estimations and RF yields. In summary, our estimations were well within the ranges of those of ORCHIDEE and LPJml.

### 3.5 Spatial distributions of the simulated yield under rainfed and irrigated conditions

Figure 9 shows the global yield distributions of *Miscanthus* and switchgrass. Under rainfed conditions, high yields are distributed in eastern US, Brazil, southern China, Africa, and Southeast Asia. To evaluate the response of yield to irrigation, we compared the results under rainfed and irrigated conditions. As shown in Fig. 9c and 9d, unconstrained irrigation greatly increased yields, especially of areas in arid regions such as the western US, southern Europe, northeastern China, India,

southern Africa, the Middle East, and coastal Australia. At the global scale, the increases (excluding the area with a polar climate) were 20.7 (from 16.8 to 37.5) Mg ha$^{-1}$ yr$^{-1}$ and 7.9 (from 7.4 to 15.3) Mg ha$^{-1}$ yr$^{-1}$ for *Miscanthus* and switchgrass, respectively, indicating that irrigated yield was more than double the rainfed yield. The spatial distributions of yield increases due to irrigation simulated by H08 were very similar to those simulated by LPJmL (Beringer et al., 2011). At the continental scale (e.g., Europe), yield increases were located mainly in southern Europe, consistent with the findings obtained using

MISCANMOD (Clifton-Brown et al., 2004). The response of yield to irrigation was weaker for switchgrass than for *Miscanthus* (see Figs. 9b and 9d). This might have been due to switchgrass having less dependency on water compared to *Miscanthus* (Mclsaac et al., 2010). *Miscanthus* growth has been reported to have a high water requirement due to its high yield, large leaf area index, and long growing season (Mclsaac et al., 2010; Lewandowski et al. 2003). As a result, *Miscanthus* yield is strongly influenced by water availability, and an annual rainfall of 762 mm yr$^{-1}$ is thought to be suitable for growth (Heaton

et al., 2019). However, the precipitation in most locations is below this level, especially in arid and semi-arid regions (see Fig. S5 in the supplementary material). Therefore, irrigation plays a critical role in ensuring optimum bioenergy crop yield in arid and semi-arid regions, especially for *Miscanthus*.

### 3.6 Effects of irrigation on yield, water consumption, and WUE in different climate zones

Climate is one of the main physical constraints of crop growth and yield. Figure 10a shows the mean yield for *Miscanthus* and switchgrass in four different Köppen climate zones (see Fig. S6 in the supplementary material). For *Miscanthus*, a tropical climate (including the northern part of South America, central Africa, Southeast Asia, and southern India) produced the highest average yield of 33.0 Mg ha$^{-1}$ yr$^{-1}$. A temperate climate (including the eastern US, Europe, southern China, and the southern part of South America) produced the second highest average yield of 19.7 Mg ha$^{-1}$ yr$^{-1}$. Dry and continental climate zones had similar average yields of 8.3 and 6.2 Mg ha$^{-1}$ yr$^{-1}$, respectively. For switchgrass, a tropical climate had the highest yield, averaging 11.9 Mg ha$^{-1}$ yr$^{-1}$. For the other three climate types, the average yields averaged 9.0, 4.7, and 4.0 Mg ha$^{-1}$ yr$^{-1}$ for the temperate, continental, and dry climate zones, respectively. As shown in Fig. 10a, irrigation greatly increased the yield, especially in dry climate zones, which had the largest yield increases of 44.2 and 15.7 Mg ha$^{-1}$ yr$^{-1}$ for *Miscanthus* and switchgrass, respectively. In contrast, irrigation had a relatively weak effect on yield in the tropical climate zone.

Figure 10b shows the water consumption for both *Miscanthus* and switchgrass. The annual mean water consumption of *Miscanthus* was around 613 mm yr$^{-1}$ for the tropical climate zone (with a high yield of 33.0 Mg ha$^{-1}$ yr$^{-1}$), whereas it was 155 mm yr$^{-1}$ for a dry climate (with a low yield of 8.3 Mg ha$^{-1}$ yr$^{-1}$) under rainfed conditions. Under irrigated conditions, the largest increases in water consumption were 1,618 and 1,054 mm yr$^{-1}$ for *Miscanthus* and switchgrass in dry climate zones, respectively. With such large amounts of irrigation, the yield in a dry climate zone can exceed that in a tropical climate zone under rainfed conditions. This highlights the yield-water tradeoff effects.

As shown in Figure 10c, the WUE values of *Miscanthus* in a tropical climate were 53.8 kg DM ha$^{-1}$ mm$^{-1}$ H$_2$O, and 53.5, 48.2, and 47.0 kg DM ha$^{-1}$ mm$^{-1}$ H$_2$O, respectively, in dry, temperate, and continental climate zones under rainfed conditions. The respective WUE values of switchgrass were 41.2, 37.9, 30.4, and 29.7 kg DM ha$^{-1}$ mm$^{-1}$ H$_2$O in continental, dry, tropical, and temperate climate zones under rainfed conditions. The WUE values for *Miscanthus* were higher than those for switchgrass, which is inconsistent with values in previous reports (VanLoocke et al., 2012). With irrigation, the WUE decreased for both *Miscanthus* and switchgrass in all climate zones. Globally, excluding the area with a polar climate, the decreases were 14.2 (from 50.6 to 36.4) kg DM ha$^{-1}$ mm$^{-1}$ H$_2$O and 12.2 (from 34.8 to 22.6) kg DM ha$^{-1}$ mm$^{-1}$ H$_2$O for *Miscanthus* and switchgrass, respectively, indicating a reduction in the mean WUE values for *Miscanthus* and switchgrass of up to 32%. This is consistent with the current global WUE trend for crops, which is high for rainfed croplands but low for irrigated croplands. However, the general magnitude of this relationship changes if the site or regional scale is considered based on reports for wheat in Syria (Oweis et al., 2000) or for wheat and maize in the North China Plain (Mo et al., 2005). Note that it might be better to use a specific crop model to investigate WUE at the site or watershed scale.

### 3.7 Improvements and limitations

Compared with earlier studies, our study made several important improvements. First, rather than using an approximation for C4 grass to represent *Miscanthus* and switchgrass in the LPJmL model, our enhanced H08 model simultaneously simulated the yields of *Miscanthus* and switchgrass at the global scale. Compared with the original H08, our enhanced model markedly decreased the mean bias (from –52% to –9% for *Miscanthus*, from 25% to –7% for switchgrass). Moreover, the growing seasons for *Miscanthus* (145–165) and switchgrass (101–114) during the period 2009–2011 at the Water Quality Field Station of the Purdue University Agronomy Center are consistent with the values of 140 and 120 reported in Trybula et al. (2015). Second, the hydrological effects of bioenergy crop production implemented in our model are actually not incorporated in some other models; for example, we considered irrigation and analyzed WUE, which was not implemented in ORCHIDEE-MICT-BIOENERGY (Li et al., 2018) and HPC-EPIC (Kang et al., 2014). Third, we investigated the differences in yield, water consumption, and WUE of both *Miscanthus* and switchgrass among different climate zones, which was useful for bioenergy land-scenario design. For example, more land can be allocated to the areas with greater WUE. In summary, our enhanced

model provides a new tool that can simultaneously simulate *Miscanthus* and switchgrass with consideration of water management (such as irrigation), although it currently considers only herbaceous bioenergy crops. From this perspective, we firmly believe that our enhanced model contributes to the bioenergy crop modelling community.

There are still several uncertainties and limitations that need to be addressed in the future. First, the bioenergy crop yield simulated by H08 did not include constraints due to nutrients, such as nitrogen and phosphorus. Nutrient dynamics are influenced by complex site-specific soil conditions (soil type, temperature, wetness, carbon, etc.), which remain quite challenging to represent properly in global models. This is why similar assumptions and limitations occur in the latest bioenergy potential/yield studies (Li et al., 2018; Yamagata et al., 2018; Wu et al., 2019). Second, the effects of $CO_2$ fertilizer and technological advancements were not considered in the current simulations. Third, our simulation was conducted with historical meteorological drivers. Therefore, yield variations in future climate scenarios under different representative concentration pathways need to be examined. Fourth, the current irrigation levels were input to represent uniform unconstrained irrigation. Further evaluations need to consider the availability of renewable water sources, and planetary boundaries of land, food, and water (Heck et al., 2018). Finally, as with other models, like MISCANMOD (Clifton-Brown et al., 2004), SWAT (Neitsch et al., 2011), and LPJml (Bondeau et al., 2007), we adopted a crop-uniform water stress formulation. However, an earlier study indicated that the water stress could be crop-specific (Hastings et al., 2009). Additional investigations of the water stress formulation for different bioenergy crops are needed.

**4 Conclusion**

In this study, we enhanced the ability of the H08 global hydrological model to simulate the yield of the dedicated second-generation herbaceous bioenergy crops. The enhanced H08 model generally performed well in simulating the yield of both *Miscanthus* and switchgrass, with the estimations being well within the range of observations and other model simulations. To the best of our knowledge, this study is the first attempt to enable a global hydrological model with consideration of water management, such as irrigation, to simulate the yield of *Miscanthus* and switchgrass separately. The enhanced model could be a good tool for the future assessments of the bioenergy–water tradeoffs. Using this tool, we quantified the effects of irrigation on yield, water consumption, and WUE for both *Miscanthus* and switchgrass in different climate zones. We found that irrigation more than doubled the yield in all areas under rainfed conditions and reduced the WUE by 32%. However, due to the low water consumption in tropical areas, the highest WUE was generally found in tropical climate zones, regardless of whether the crop was irrigated.

*Code and data availability.* The code of the model used in this study is archived on Zenodo (https://zenodo.org/record/3521407#.XbjZqiXTZMB) under the Creative Commons Attribution 4.0 International License. Technical information about the H08 model and the input dataset are available from the following website: http://h08.nies.go.jp.

*Competing interests.* The authors declare that they have no conflict of interest.

*Author contribution.* Naota Hanasaki designed this study. Zhipin Ai collected the data, developed the model code, and performed the simulations. Zhipin Ai prepared the manuscript, with contributions and comments from Naota Hanasaki, Vera Heck, Tomoko Hasegawa, and Shinichiro Fujimori.

*Acknowledgments.* This study was supported by the Environment Research and Technology Development Fund (JPMEERF20202005 and JPMEERF15S11418) of the Environmental Restoration and Conservation Agency, Japan.

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

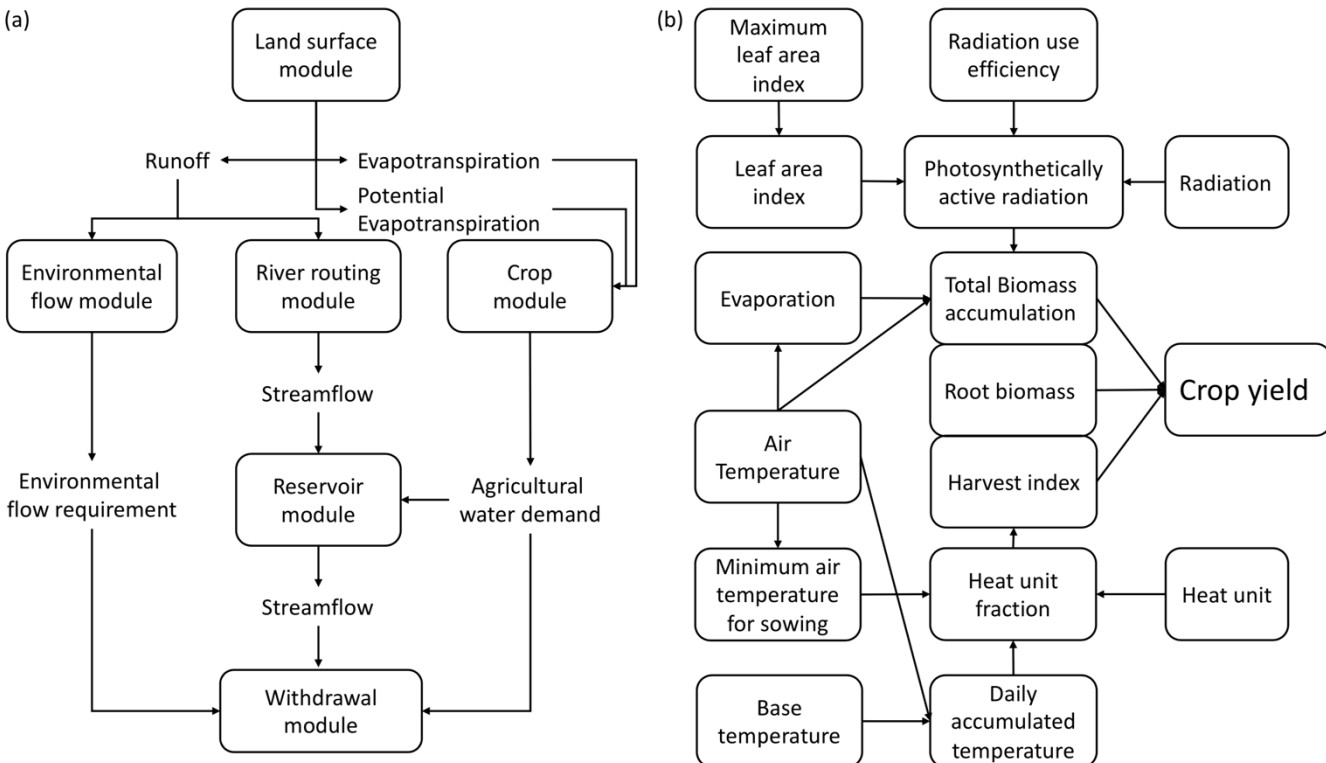

**Fig. 1 Schematic diagram showing the six submodules (a) and basic biophysical processes of the crop module (b) in the H08 model.**

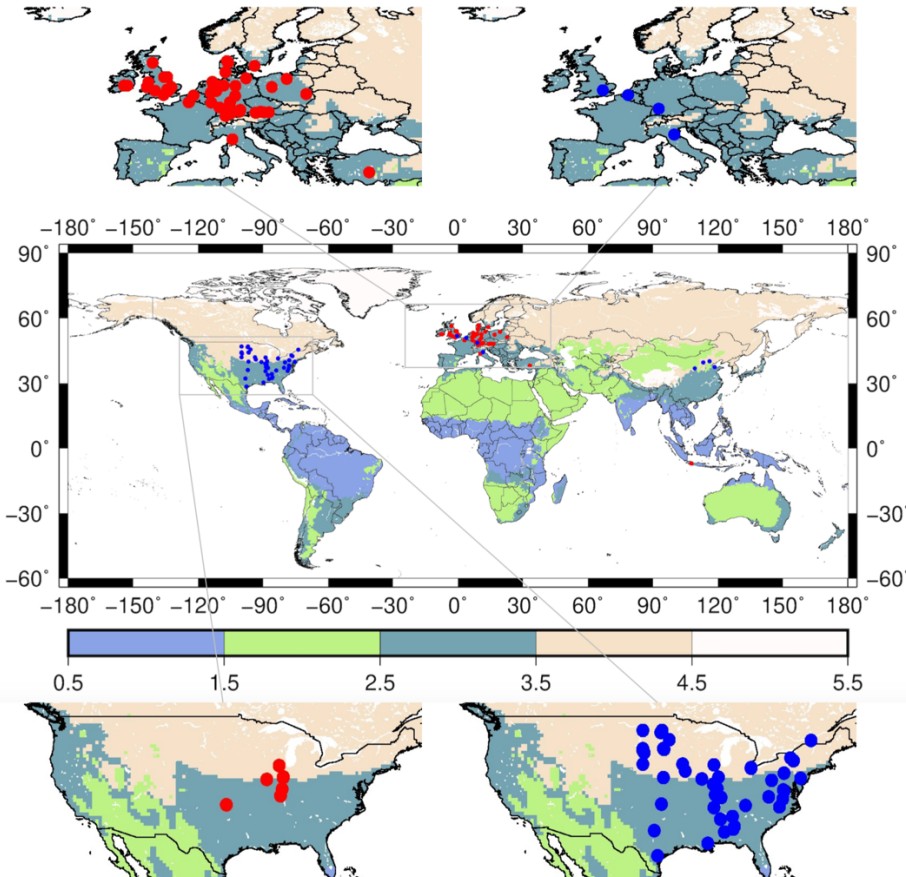

**Fig. 2 Map showing the locations of the *Miscanthus* (red dots) and switchgrass (blue dots) sites under rainfed condition and the Köppen climate zones. The specific categories are the 1 (light blue) tropical, 2 (light green) dry, 3 (light teal) temperate, 4 (light tan) continental, and 5 (light peach) polar climate zones.**

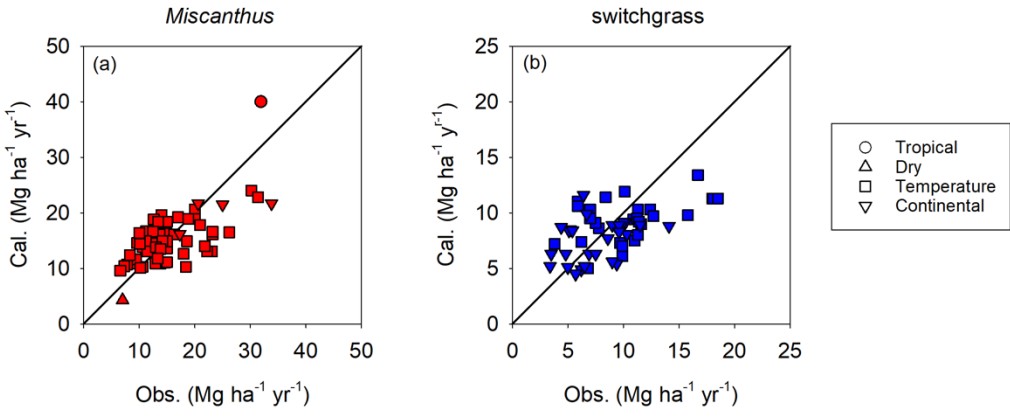

**Fig. 3 Overall comparison of the calibrated (Cal.) and observed (Obs.) yields for *Miscanthus* and switchgrass. The black line is the 1:1 line.**

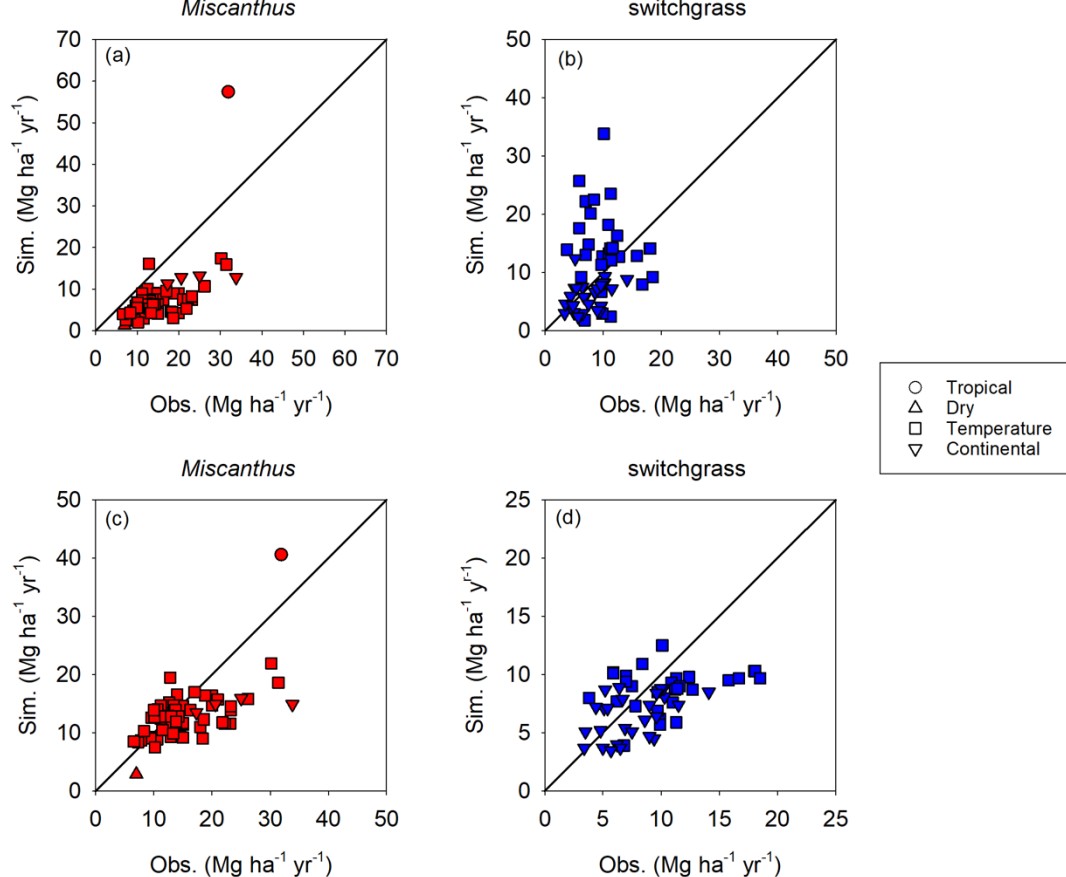

**Fig. 4 Overall comparison of the simulated (Sim.) and observed (Obs.) yields for *Miscanthus* and switchgrass. The simulated yields in (a) and (b) are from the original H08 model, whereas those in (c) and (d) are from the enhanced H08 model. The black line is the 1:1 line.**

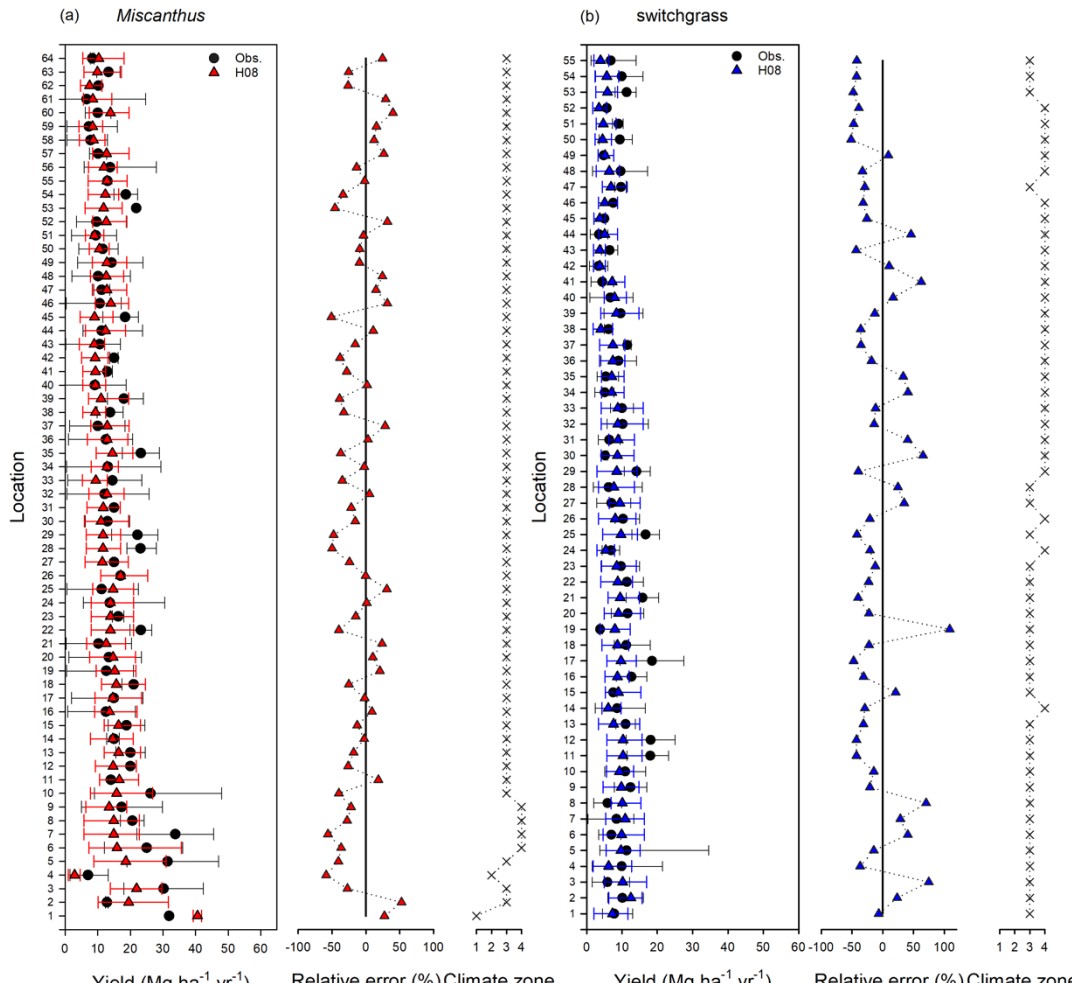

**Fig. 5 Site-specific performance (presented with latitude increasing from the bottom of the vertical axis) and relative error of the simulated yield obtained using the enhanced H08 model compared with the observed yields for *Miscanthus* and switchgrass. The longitude and latitude of each location for *Miscanthus* and switchgrass are given in Tables S1 and S2, respectively. The thin "x" indicates the site's climate, where 1, 2, 3, and 4 refer to the tropical, dry, temperate, and continental climate zones, respectively. Obs. means the observed mean yield. The black error bar represents the range of the observed minimum and maximum yield. The red or blue error bar represents the range of the simulated minimum and maximum yield.**

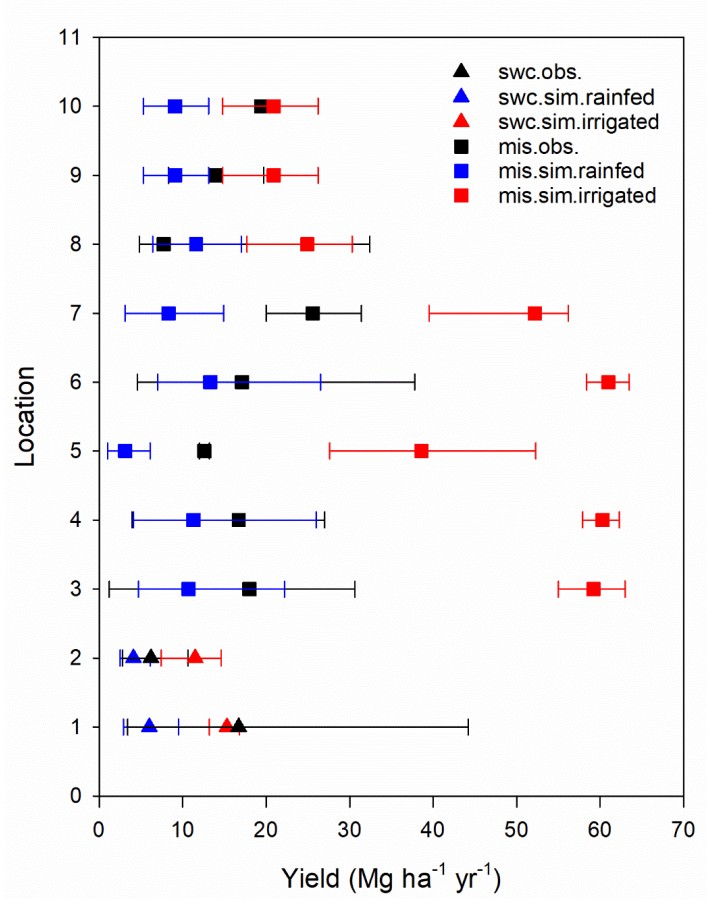

**Fig. 6 Site-specific performance (shown with increasing latitude from the bottom of the vertical axis) of the simulated yield (sim.) obtained using the enhanced H08 model compared with observed yields (obs.) for *Miscanthus* (mis.) and switchgrass (swc.) under irrigated condition. The longitude and latitude of each location ID for *Miscanthus* and switchgrass are given in Table S3. Obs. indicates the observed mean yield. The black error bar represents the range of the observed minimum and maximum yield. The red or blue error bar represents the range of the simulated minimum and maximum yield.**

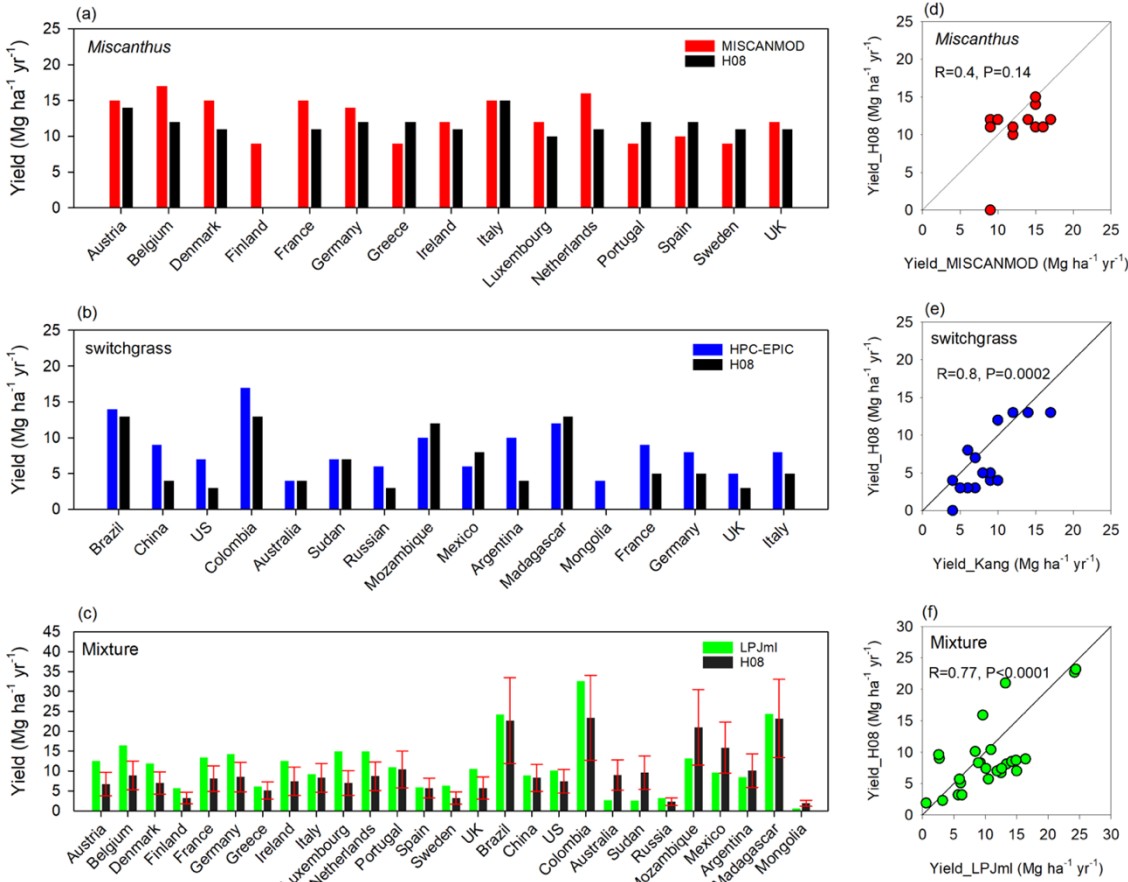

**Fig. 7 An independent country-specific comparison of the yield simulated by the enhanced H08 model with those of three other models (MISCANMOD, HPC-EPIC, and LPJmL) for *Miscanthus* (a, d), switchgrass (b, e), and their combination (c, f). The H08 in (c, f) indicates the average yield of *Miscanthus* and switchgrass, and the upper and lower error bars in (c) represent the yields for *Miscanthus* and switchgrass, respectively.**

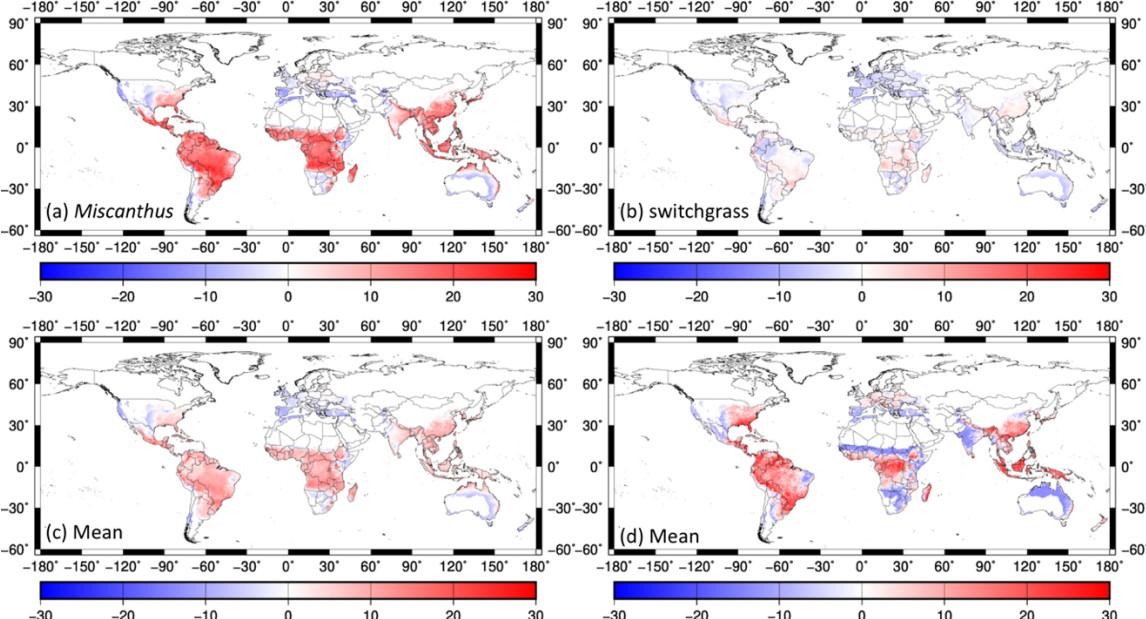

**Fig. 8 Comparison of the yield difference (simulated yields minus RF yields) between model simulations and the RF map (Li et al., 2020): a) for *Miscanthus* with the yield from H08 minus that from RF, b) for *switchgrass* with the yield from H08 minus that from RF, c) for the mean of *Miscanthus* and switchgrass with the yield from H08 minus that from RF, d) the ensemble yield of *Miscanthus* and switchgrass with the yield from LPJml minus that from RF.**

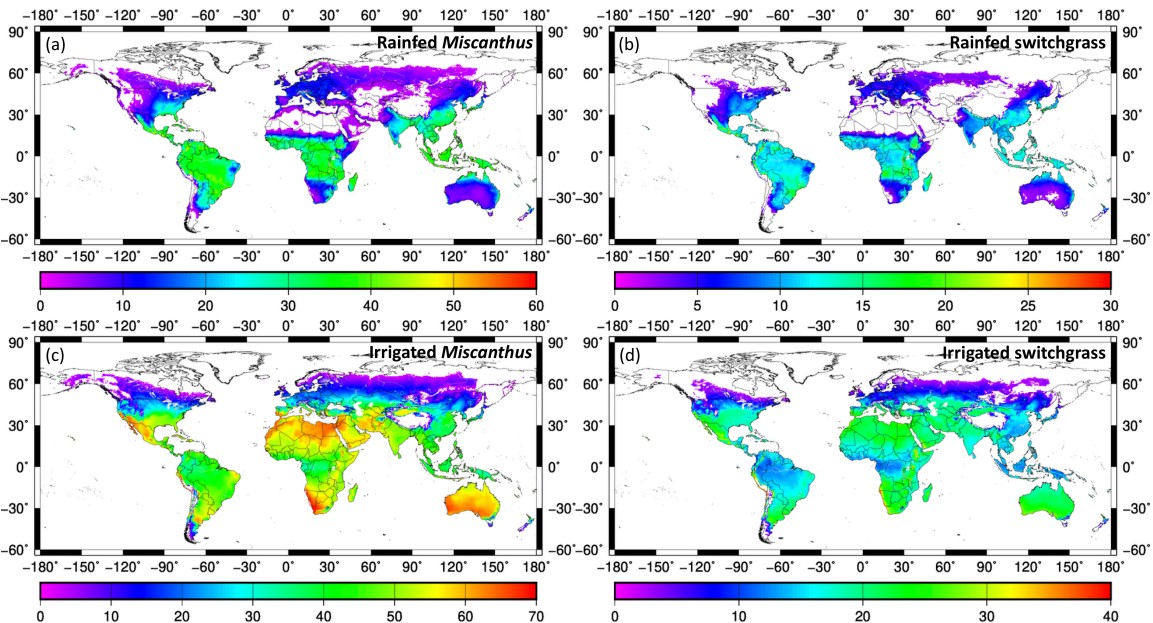

**Fig. 9 Spatial distributions of the simulated yields (exceeds 2 Mg ha⁻¹ yr⁻¹) for *Miscanthus* (a, c) and switchgrass (b, d)**

**under rainfed (a, b) and irrigated (c, d) conditions. The units for the legends are Mg ha⁻¹ yr⁻¹.**

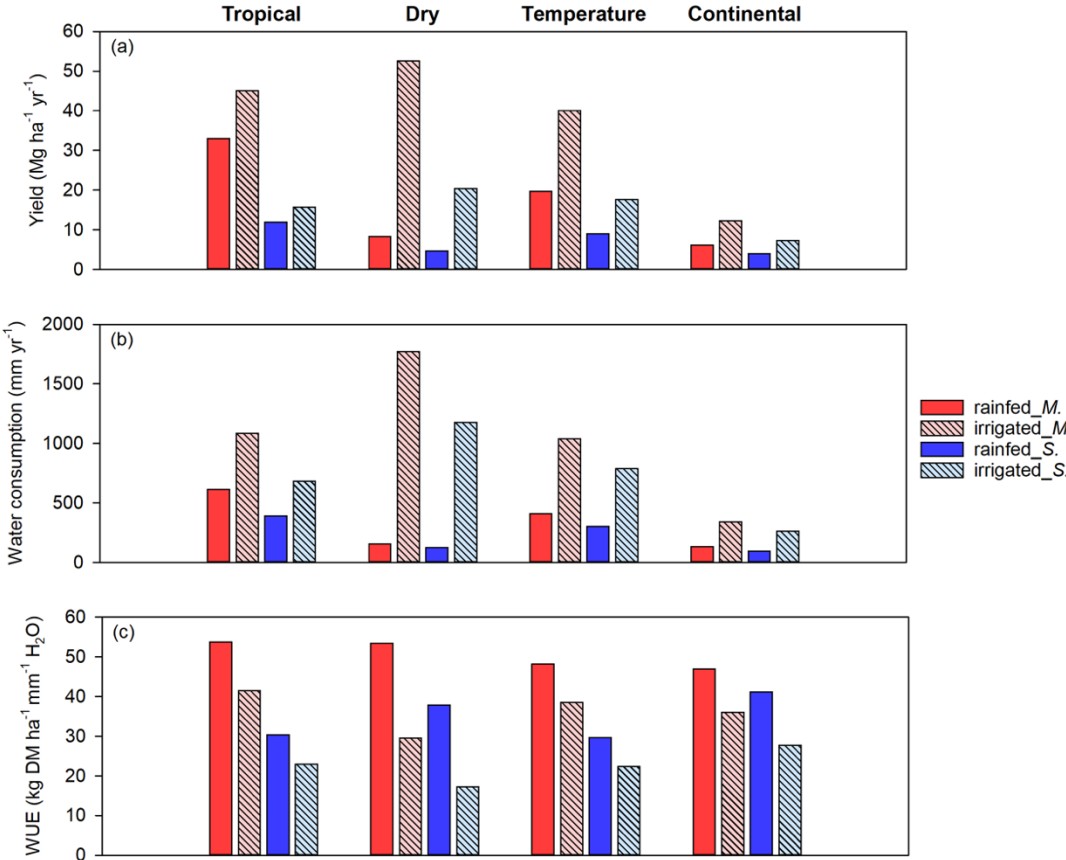

**Fig. 10 Variation in the average yield (a), crop water consumption (b), and water use efficiency (WUE) (c) of *Miscanthus* and switchgrass under rainfed and irrigated conditions in four different Köppen climate zones (tropical, dry, temperate, and continental climates) based on meteorology data collected from 1979 to 2016. The abbreviations *M.* and *S.* in the legend denote *Miscanthus* and switchgrass, respectively.**

**Table 1. Parameter abbreviations and explanation**

| Parameter abbreviation | Full name | Physical meaning |
| --- | --- | --- |
| Hun | Potential heat unit | The value of potential heat units required for the maturity of the crop |
| be | Radiation use efficiency | The potential growth rate per unit of intercepted photosynthetically active radiation |
| To | Optimum temperature | The optimal temperature for plant growth |
| Tb | Base temperature | The base temperature for plant growth |
| blai | Maximum leaf area index | The maximum potential leaf area index |
| dlai | Fraction of growing season when growth declines | Same as the full name |
| dpl1 | Complex number1 | First point on the optimal leaf area development curve. Before decimal: fraction of growing season, after decimal: max corresponding LAI. |
| dpl2 | Complex number2 | Before decimal: fraction of growing season, after decimal: max corresponding LAI. Second point on the optimal leaf area development curve. |
| rdmx | Maximum Rooting Depth | Same as the full name |
| Hunmax | Maximum daily accumulation of temperature | Same as the full name |
| TSAW | Minimum temperature for planting | Same as the full name |

**Table 2. Parameters set in the enhanced H08 model.**

| Bioenergy crop | Parameter | Old value | New value | Source |
|---|---|---|---|---|
| *Miscanthus* | Hun | 9999 | 1,830 | Trybula et al., (2015) |
| | be | 39 | 38 | Calibrated |
| | To | 30 | 25 | Trybula et al., (2015); Hastings et al.,(2009) |
| | Tb | 10 | 8 | Calibrated |
| | blai | 11.5 | 11 | Calibrated |
| | dlai | 0.85 | 1.1 | Trybula et al., (2015) |
| | dpl1 | 10.2 | 10.1 | Trybula et al., (2015) |
| | dpl2 | 50.95 | 45.85 | Trybula et al., (2015) |
| | rdmx | 4 | 3 | Trybula et al., (2015) |
| | Hunmax | 12.5 | 11.5 | Calibrated |
| | TSAW | 10.0 | 8.0 | Calibrated |
| Switchgrass | Hun | 9999 | 1,400 | Trybula et al., (2015) |
| | be | 47 | 22 | Calibrated |
| | To | 25 | 25 | Trybula et al., (2015) |
| | Tb | 12 | 10 | Calibrated |
| | blai | 6 | 8 | Calibrated |
| | dlai | 0.7 | 1 | Trybula et al., (2015) |
| | dpl1 | 10.2 | 10.1 | Trybula et al., (2015) |
| | dpl2 | 20.95 | 40.85 | Trybula et al., (2015) |
| | rdmx | 2.2 | 3 | Trybula et al., (2015) |
| | Hunmax | 12.5 | 15.5 | Calibrated |
| | TSAW | 10.0 | 8.0 | Calibrated |

**Table 3. Parameter ranges for the calibration simulations.**

| Bioenergy crop | Parameter | Range | Increment | Unit | Reference |
|---|---|---|---|---|---|
| *Miscanthus* | be | (30, 40) | 2 | g MJ$^{-1}$ × 10 | Clifton-Brown et al., (2000); van der Werf et al., (1992); Beale and Long, (1995); Heaton et al., (2008); Trybula et al., (2015) |
| | blai | (9, 11) | 1 | m$^2$ m$^{-2}$ | Heaton et al., (2008); Trybula et al., (2015) |
| | Tb | (7, 9) | 1 | °C | Beale et al., (1996); Trybula et al., (2015) |
| | Hunmax | (11.5, 16.5) | 1 | °C | H08 Endogenous variable |
| | TSAW | (8, 10) | 1 | °C | H08 Endogenous variable |
| Switchgrass | be | (12, 22) | 2 | g MJ$^{-1}$ × 10 | Heaton et al., (2008); Madakadze et al., (1998); Trybula et al., (2015) |
| | blai | (6, 8) | 1 | m$^2$ m$^{-2}$ | Trybula et al., (2015); Giannoulis et al., (2016); Madakadze et al., (1998); Heaton et al., (2008) |
| | Tb | (8, 10) | 1 | °C | Trybula et al., (2015) |
| | Hunmax | (11.5, 16.5) | 1 | °C | H08 Endogenous variable |
| | TSAW | (8, 10) | 1 | °C | H08 Endogenous variable |