# Peer review of "Simulating second-generation herbaceous bioenergy crop yield using the global hydrological model H08 (v.bio1)"

_Geoscientific Model Development, 2020_

## Referee Comment (RC1) · Anonymous Referee #1 · 16 Aug 2020

The paper discusses some enhancements to the H08 global hydrologic model to simulating bioenergy yield over a history. The authors compare the results to previous assessments and some observed yield values around the globe. The paper is a good contribution, and I recommend its publication. However, the paper has several sections that require some additional clarity/details. Below I provide a detailed summary of some of these issues.

- It is not clear how this work builds on previous work by the authors (Yamagata et al. 2018 and Wu et al. 2019) or by the work of Trybula et al 2015.
- Better documentation of the methodology section to allow for reproducibility including the equations, and the explanation of the various parameters. A schematic would be also help.
- Sections 2.1 and 2.2 leave the reader wondering about the specifics of the two-step approach discussed, and how the adopted enhancements build on the previous approach. These two sections deserve more details of the methodology with greater levels of details that what is being offered. This will help the reader understand exactly how this work differs and builds on the two previous studies by the team, how to interpret the results and the difference between the 'original' and 'enhanced' versions of H08 (figure 3), how to interpret the various variables shown in Table 1, and to facilitate reproducibility.
- The paper shows some validation results for the rainfed module, and not for the irrigation module, but then show results for both when simulating both globally. The validation step for the irrigated module should be shown and discussed in the main text.

Other issues:

I would suggest shortening the title. How about something like "Simulating second-generation bioenergy crop yield using the global hydrologic model H08"

Line 4. Why is Miscanthus capitalized and italic but not switchgrass?

Line 7: 'enhanced H08' Doesn't H08 keep track of different version numbers that can be used here instead of calling something an enhanced model version?

Line 13: Add a sentence into the abstract to introduce the term BECCS if you are going to start the introduction section with this term. Preferably, I would suggest confining the framing around bioenergy crops rather than BECCS since the latter term never appears again in the text.

Lines 26, 30, 34: LPLmL should be LPJmL

Lines 30-32: It is not just LPJmL based on the following paragraph. It is also H08 based on the two recent publications using H08 (Yamagata et al. 2018 and Wu et al. 2019).

Line 34: change 'biogeny' to bioenergy

Line 41: Hanasaki et al 2008a/b are repeated twice in the list.

Line 44: the reference Wu et al. 2019 is missing in the list of references at the end.

Lines 49-50: I would suggest omitting the sentence "However, it is noted that the model performance for the simulated bioenergy crop yield was not validated at all" as an argument to justify the novelty of the work. I doubt the authors are claiming that the previous two studies using H08 with representation of bioenergy yield ignored properly validating the model and this study contributes this novelty. I would

suggest that authors replace this sentence with an explanation of how the new work builds on the two-step approach documented in the two previous papers (Yamagata et al. 2018 and Wu et al. 2019).

Line 61: 'The six sub-modules', You have not introduced what those six submodules are yet. I would start by listing them or at least list them in () right after this phrase.

Line 75: I would expand on this section to show the two-step approach here before talking about model enhancements in the next section (2.2). Even if those were presented in the two previous publications, I would at least include them in SI to make this manuscript a standalone piece.

Lines 76-85: I would suggest including all the equations and steps for how yield is simulated to shed more light about the method and to allow reproducibility of the approach.

Line 90: 'as an output item' Are you saying that can you simulate water consumption as a new output variable? It is not clear.

Line 91: 'Fifth, we fixed the bug in the original code'. What Bug? One could say 'we fixed a bug in the original code'. But this is so vague and does not really give the reader any additional information. I would suggest dropping the fifth point. Such details are best documented in SI.

Lines 105-110: can you mention the number of data points and years being used?

Line 115: what variable is being calibrated here? H08 simulates many output variables. How does the calibration process ensure that the adopted calibration process does not offer a gain in better matching one variable at the expense of another variable? For example, did the authors calibrate runoff first and then yield, or is it done all at once? If it is the latter, then showing some results on runoff would be necessary. I am not asking the authors to necessarily do additional work, but rather to better explain their approach.

Line 117: 'the enhanced h08'. Does this mean that the second simulation was only done for the enhanced model?

Lines 124-125: A bit unclear. Was the calibration done as a multi-objective optimization process to optimize both the RMSE and R values.  For example, how do you decide an optimal parameter set when the two goodness-of-fit variables disagree? Figure 3 only shows RMSE, so I would suggest that you stick to this one and drop the R coefficient. Also, it is not clear if observed data is available for several individual years or only a single average year is available. If a time series exists, then I would suggest using goodness-of-fit measures such as Nash-Sutcliffe.

Line 137: 'because the few sites that were irrigated'. Please rephrase.

Line 139: 'previous reports' Please add citations to support this claim. The single sentence that comes afterward is insufficient. What about other parameters?

Lines 140-145: how does this work differ from Trybula et al 2015? This is not discussed in the intro. Also given that the adopted approach follows the SWAT implementation in Trybula et al 2015, and almost all of the parameters taken from the literature are also taken from Trybula et al 2015, would not it guarantee that you get similar parameter values for the other calibrated values to match those in Trybula et al 2015? What about other studies?

Lines 148-154: although the results are better than the original version, the results still seem to show a tendency to underestimate based on the results shown in figure 3.

Line 158: 'well at sites 1, 2, and 10' so how many sites are under irrigation? You should mention it here.

Lines 166-175: Did you drop the missing value from the significance test analysis (e.g., Finland in Fig 5d, Mongolia in Fig 5e)? I am still unsure whether the yield values from the other studies are average values over a particular period, and if it is the same period as in this study.

Lines 188-189: 'This can also be inferred from the validation results in Heck et al. (2016)' Please elaborate.

Lines 196-201: This information should appear earlier in the manuscript, so the reader is left wondering about such details. Also, if there is annual data from the other studies, then why not look at the timeseries instead of simply comparing the average value over a time period? To say a model can capture the long term mean over different basins is one level of validation, but to say that the model can also capture the interannual variability of yield from year to year, then this is a much more desirable level of validation.

Lines 203-220: This section comes as s surprise as it was not mentioned earlier as part of the framing of the paper in the front sections.

Line 206: It is confusing how many simulations were done in the study. The authors talk about two simulations twice, but are referring to different ones. I would suggest including an experimental design section as part of the methodology section to explain the different simulations to be conducted over a historical period (rainfed/irrigated, original/enhanced, …).

Lines 211-220: The validation results shown and discussed in the main body of the manuscript only talk about the rainfed simulations. It is unconvincing to skip the validation step for the irrigation module, and then show results and draw conclusions using that irrigation modeling capability. In this section, results from this study are shown, but they are not contrasted with estimates from previous studies.

Lines 223-233: Is this based on some aggregated regions, or on all the grids that belong to each climate zone? How do these results compare to other studies that were discussed in section 3.4? assuming this was based on a grid-level analysis, why not plot the results for all the grids and show a scatter plot (yield on the y-axis, and aridity or some other index that allows for distinguishing among the different climate zones on the x-axis)? This would allow the authors to fit a line to the data and talk about the results in a more compelling way.

Line 241: 'WUE, which is defined in this study as the ratio of yield to water consumption'  This should have appeared the first time the term is mentioned in the manuscript.

Line 246: 'The WUE values for Miscanthus were higher than those for switchgrass, which is inconsistent with values in previous reports' Please add a sentence to articulate why?

Line 263: 'which was useful for optimizing bioenergy land with better consideration of water protection' – I am not sure what this means?

Lines 266: 'and our results are reproducible with the transparent parameter disclosed.' Just sharing the parameters sets does not guarantee reproducibility. I would suggest omitting that phrase.

Line 277: why was not this yield map used in the previous sections as part of the validation exercise? Also, I would suggest moving figure S7 out of SI and into the main text.

Figure 4: To be consistent with the black error bars, the blue/red ones should also reflect max/min. Also, why include all the years for observations? Should not these be for the years for which there is an associated observed yield value?

Figure 7: why is the y-axis for panel b flipped around as if the values should be negative? I would suggest keeping it consistent with the other two panels (0 at the bottom left corner, and the bar chart goes upward for positive values).

Table 1: Please add another column to define the different parameters and what they mean physically.

---

## Referee Comment (RC2) · Anonymous Referee #2 · 24 Aug 2020

This manuscript enhanced the capability of a global hydrological model named H08 in simulating two perennial bioenergy crops, Miscanthus and switchgrass. The results were validated against site-level and country-level observed crop yields. The enhanced model is applied to simulate the impact of irrigation on crop water consumption and water use efficiency compared to rainfed condition. This study makes contribution to study the impact of large-scale deployment of bioenergy crops on water resources. However, I have some major comments as listed below.

General comments: 1. Model validation: This study only validates the simulated yield

results against observations for Miscanthus and switchgrass. While the main contribution/innovation of this study is on hydrological applications, this study didn't validate any variables for the water cycle, including evapotranspiration, runoff, and irrigation. Without such validations, I feel difficult to be convinced for the reliability of the simulated results for crop water consumption and WUE.

2. Study innovation: The Introduction didn't well motivate the study and present the novelty/uniqueness of this study. For example, the argument "However, it is noted that the model performance for the simulated bioenergy crop yield was not validated at all." is a little bit difficult to be taken as an innovation of this study. And almost all the parameter values were directly taken from Trybula et al. 2015, which makes me wonder what are the main differences/improvements of this current study compared to Trybula et al. 2015? Given the difference between H08 used in this study and SWAT used in Trybula et al. 2015, can the authors justify the applicability of directly using SWAT's parameter values?

3. Model description: This study only describes the crop module in H08 without much descriptions for the hydrological module in the model, especially given the important role of hydrological processes in this study. In addition, many indices and simulations (e.g., using new meteorological dataset) were not well described in the methods section, such as how WUE is calculated, how irrigation works, and how many simulations were conducted in total and their respective purposes.

4. Paper organization: The main context is missing many important information (e.g., sensitivity test results, model descriptions, equations). Many important information and results were given in the SI rather than directly presented in the main context. The methods section is missing descriptions for the simulations conducted in this study and many new simulations came out suddenly in the Results sections. It will be necessary to reorganize the paper and move some important information from SI to the main context.

5. Limitation in discussion: the current results and discussions are quite limited. For example, quantitative evaluations for model improvements were missing. What are the improvements of the enhanced H08 compared to its old version which uses C4 grass to characterize switchgrass and Miscanthus? One of the most important features of switchgrass and Miscanthus is their perennial features and longer growing seasons, but this study didn't have any discussions on this kind of perspectives.

Specific comments: Lines 31-36: Actually, CLM5 also has the irrigation scheme and river routing and CLM5 also includes both bioenergy crops and the water cycle.

Line 34: typo, should be "bioenergy and the water cycle"

Line 61: I am curious does it mean H08 can only simulate hydrological processes and crop growth as a 0.5 degree and at a daily scale? How about other spatial and temporal resolutions? Can H08 simulate GPP and LAI? If so, how about the simulation results for GPP and LAI?

Lines 61-64: What are the six sub-modules? It will be great if the authors can add more descriptions for the H08 model (e.g., calculations/illustrations for the hydraulic processes), as not every reader is familiar with H08.

Line 75: what is single-irrigated and double-irrigated mean?

Line 85: what is "substantially" mean? Can you quantify the changes?

Section 2.2: Can you change some descriptions into equations? For example, how did you calculate the output item for water consumption and WUE? If they are already in the supplementary materials, it will be great if you can move some key equations to the main context. What is the bug in the original code?

Section 2.5: How is irrigation calculated in H08, such as the irrigated area and irrigated amount?

Line 23 under section 2.5: since 1944 simulations were conducted, can you give more

results for the ensemble runs rather than just present the one with lowest RMSE? For example, what are the uncertainty ranges for the calibrations? What are the sensitivity results for all the calibrated parameters? Here the authors only mentioned the most sensitive parameter names in line 20 but no results were given to support it.

Line 38 in section 3.1: change to "because only few sites were irrigated".

Line 38 in section 3.1: can you add reference after the "previous reports"?

Line 50 in section 3.2: can the authors add more quantitative results and discuss the reasons/mechanisms for why the over- and under-estimations have been addressed in the enhanced H08? Actually Miscanthus is still underestimated in the enhanced H08, why?

Line 58 in section 3.2: what are sites 1, 2, and 10? Can you refer to more specific names or descriptions for those sites, as these site numbers are not quite meaningful?

Line 59 in section 3.2: again, adding irrigation scheme in H08 in the methods section will be helpful.

Line 64 in section 3.2: the two results were similar, but what are the implications? What are the differences between the two meteorological datasets? Also, it makes me wonder how many simulations or how many kinds of simulations were conducted in this study? This new simulation with additional meteorological dataset never mentioned in the methods section. I will suggest the author add a new table or at least a new paragraph in the methods section to better illustrate the simulations conducted in this study, including their names, descriptions, differences, purposes, etc.

Section 3.3: can you add those correlation and significant level values in Figure 5 as well?

Line 10 in section 3.4: grammar error for the sentence

Line 55-58 in section 3.6: again, how is the current results compared to old H08 which

uses C4 grass to represent Miscanthus and switchgrass?

Line 63-65 in section 3.6: I doubt the argument that the enhanced H08 is the only model that can simultaneously simulate Miscanthus and switchgrass with consideration of water management, as CLM5 also has this capability.

Tables 1 and 2: could you add the long name or descriptions for these parameters? What is "step" mean in Table 2?

Figure 1: could you add a flow chart or schematic figure for the hydrological processes in H08 or the overall model structure?

Figure 3: can the authors decrease the maximum magnitudes for figure b and d, like to be 40, since no data exceeds 40 and right now most of the points are centered to a very small range? And can the authors add a third axis (e.g., different colors) to distinguish the locations/climate zones for the points?

Figure 6: it will be helpful to add a title name in the figure, e.g., (a) Rainfed Miscanthus.

Figure 7: it may be helpful to move Figure S6 to the main context and combined with Figure 2 to better illustrate the methods section. But the authors can decide after revise the methods section.

---

## Author Response (AR1)

Dear Dr. Hisashi Sato,

Thank you very much for taking time to handle our paper (gmd-2020-179). We are grateful to the Reviewers for their valuable comments and suggestions. We have the pleasure of enclosing a revised version of the manuscript titled "Simulating second-generation herbaceous bioenergy crop yield using the global hydrological model H08 (v.bio1)" and a detailed response to the Reviewers' comments below. We hope that the revised manuscript has been strengthened for publication in *Geoscientific Model Development*.

In the responses below, we have addressed each of the Reviewers' comments in detail. The comments from each Reviewer are noted as "R" (e.g., R1) while each comment is noted as "C" (e.g., C1) to better index all comments. The line numbers indicated refer to the revised manuscript (clear version). In the manuscript (with track changes), the updates are colored in red and the deletions are strikethrough in blue. We hope the manuscript is now suitable for publication in *Geoscientific Model Development*. Please do not hesitate to contact us if you require any further information.

Sincerely yours,
Zhipin Ai (on behalf of co-authors)

**Response to Reviewer 1**

| Response to Reviewer 1 | | |
|---|---|---|
| No. | Comment | Response |
| R1C1 | The paper discusses some enhancements to the H08 global hydrologic model to simulating bioenergy yield over a history. The authors compare the results to previous assessments and some observed yield values around the globe. The paper is a good contribution, and I recommend its publication. However, the paper has several sections that require some additional clarity/details. Below I provide a detailed summary of some of these issues. | Deer Reviewer, thank you very much for taking time to carefully read our manuscript. We are pleased to see your recommendation for publication. Your valuable comments enabled us to clarify a number of points that we previously unaware of, and we hope that we have increased the quality of the manuscript substantially. We have revised the paper by trying to incorporate all relevant comments and remarks. We have also tried to respond to all the comments meticulously as you may see below. Please find our responses to each comment below. |
| R1C2 | It is not clear how this work builds on previous work by the authors (Yamagata et al. 2018 and Wu et al. 2019) or by the work of Trybula et al 2015. | We apologize for the unclear description. Here, let us further explain how our work builds on that of Yamagata et al. (2018), Wu et al. (2019), and Trybula et al. (2015). In the whole, the first bioenergy crop implementation in H08 was conducted by Yamagata et al. (2018). Using outputs from the same model employed by Yamagata et al. (2018), Wu et al. (2019) predicted future global bioenergy potential. Our study is a substantial upgrade to the portion of Yamagata et al. (2018) purely dedicated to the improvement of bioenergy crop modeling. In this upgrade, we referred the parameters reported by Trybula et al. (2015), which provided crop parameters for the leaf area development curve.

To be specifically, in the work of Yamagata et al. (2018), the bioenergy crop modeling was realized in two steps. First, crop parameters (see the old values in Table 2) for *Miscanthus* (refer to *Miscanthus giganteus* in this study) and switchgrass (refer to *Panicum virgatum* in this study) |

| | | | were adopted based on the settings of the SWAT model 2012 version (Arnold et al., 2013). However, the default parameters could not well reflect the characteristics for *Miscanthus* and switchgrass and could lead to serious bias based on the result in Trybula et al. (2015). Second, because both *Miscanthus* and switchgrass are perennial, the potential heat unit was set as unlimited (see the old values in Table 2). However, this potential heat unit is far from the observations reported by Trybula et al. (2015) (see the new values in Table 2). Here, further enhancements were therefore conducted as follows. First, we changed the leaf area development curve by adopting the potential heat unit (Hun) and leaf area related parameters (dpl1 and dpl2) proposed by Trybula et al. (2015). The potential heat unit can determine both the total cropping days and the leaf development. Here, we set it at 1,830 and 1,400 degrees for Miscanthus and switchgrass, respectively, as recommended by Trybula et al. (2015) based on their field observations. This modification changed the original heat unit index (Ihun) and the development of the leaf area index curve. Second, we modified the algorithm for water stress that was used to regulate the radiation use efficiency. We took the ratio of actual evapotranspiration to potential evapotranspiration as the water stress factor for any point in the simulation, similar to the description of the soil moisture deficit used in other studies (Anderson et al., 2007; Yao et al., 2010). Third and the most important, we conducted a systematic parameter calibration and evaluation for both *Miscanthus* and switchgrass with the best available data. |
|---|---|---|---|
| R1C3 | | Better documentation of the methodology section to allow for reproducibility including | Thank you. We have added the equations related to yield estimation to Section 2.1; |

| | | the equations, and the explanation of the various parameters. A schematic would be also help. | added an explanation of the parameters in Table 1, and provided the original values of the parameters in Table 2. In addition, we have described the original implementation of bioenergy crops in Section 2.2; rephrased the calibration process in Section 2.2; revised Fig.1 to include both the submodules of H08 and the specific biophysical processes of crop module; and improved Fig. 2 by adding the climate zone information originally presented in Fig. S6 to better illustrate the site locations. |
|---|---|---|---|
| R1C4 | | Sections 2.1 and 2.2 leave the reader wondering about the specifics of the two-step approach discussed, and how the adopted enhancements build on the previous approach. These two sections deserve more details of the methodology with greater levels of details that what is being offered. This will help the reader understand exactly how this work differs and builds on the two previous studies by the team, how to interpret the results and the difference between the 'original' and 'enhanced' versions of H08 (figure 3), how to interpret the various variables shown in Table 1, and to facilitate reproducibility. | We agree with the Reviewer's concern and have largely revised the Sections 2.1 and 2.2. The main modifications are as follows: we added the most important equations used for crop yield modeling to Section 2.1; we revised Section 2.2 to illustrate the original implementation of the bioenergy crop (two-step approach) in H08 and our enhancement; we included the original parameter settings in Table 2 and the physical meanings of the parameters in Table 1; and we clarified the six submodules of H08 in Section 2.1 and revised Fig.1 by adding a schematic diagram of the connections for each submodule. |
| R1C5 | | The paper shows some validation results for the rainfed module, and not for the irrigation module, but then show results for both when simulating both globally. The validation step for the irrigated module should be shown and discussed in the main text. | Thank you for this good suggestion. We have moved the validation results (site-level) with irrigation in Fig. 6. The main text has been edited as follows (lines 240–246):

 "We also investigated the performance under the irrigated condition (shown in Fig. 6). We used the reported observed yields for ten sites globally (Table S3). We found that the simulated yields were within or close to the observed yields for five sites located in China, the UK, and France (see Table S3), but were overestimated for the remaining sites. This was due to the assumption of irrigation. H08 assumes that irrigation is fully applied to crops |

| | | and hence the yield represents the maximum potential yield under irrigation condition. Therefore, if the reported yield is within the range of the simulated yield between rainfed and irrigated conditions, it is considered reasonable. This was found to be the case, as shown in Fig. 6." |
| --- | --- | --- |
| | | We also included the validation results with irrigation (country-level) in Fig. S3, which indicates good performance. The corresponding text is on lines 268–270: |
| | | "An additional comparison under the irrigated condition is presented in Fig. S3. The correlation coefficient of the yield simulated by H08 and LPJmL, as shown in the scatterplot (Fig. S3), was 0.95. A t-test showed that the correlation was significant at the 0.01 level." |
| R1C6 | I would suggest shortening the title. How about something like "Simulating second-generation bioenergy crop yield using the global hydrologic model H08" | We have shortened the title to "Simulating second-generation herbaceous bioenergy crop yield using the global hydrological model H08 (v.bio1)". |
| R1C7 | Line 4. Why is Miscanthus capitalized and italic but not switchgrass? | *Miscanthus* denotes *Miscanthus giganteus* and switchgrass indicates *Panicum virgatum* in this study. *Miscanthus* is the genus to which the studied species belongs, which is always capitalized and italicized in Binomial nomenclature. Therefore, we have used this conventional expression (capitalized and italicized) for *Miscanthus*. The same expression has been used in previous reports such as Trybula et al. (2015). |
| R1C8 | Line 7: 'enhanced H08' Doesn't H08 keep track of different version numbers that can be used here instead of calling something an enhanced model version? | Thank you. We have changed 'enhanced H08' to 'H08 (v.bio1)'. |
| R1C9 | Line 13: Add a sentence into the abstract to introduce the term BECCS if you are going to start the introduction section with this term. Preferably, I would suggest confining the framing around bioenergy crops rather | Thank you for noting this issue that we previously unaware of. We have taken your suggestion to focus on the bioenergy crop plantation and removed the abbreviation of BECCS. |

| | than BECCS since the latter term never appears again in the text. | |
|---|---|---|
| R1C10 | Lines 26, 30, 34: LPLmL should be LPJmL | We have corrected this error. |
| R1C11 | Lines 30-32: It is not just LPJmL based on the following paragraph. It is also H08 based on the two recent publications using H08 (Yamagata et al. 2018 and Wu et al. 2019). | Thank you. We have added H08 here. |
| R1C12 | Line 34: change 'biogeny' to bioenergy | We have corrected this error. |
| R1C13 | Line 41: Hanasaki et al 2008a/b are repeated twice in the list. | This citation is listed as "(Hanasaki et al., 2008a, 2008b, 2010, 2018a, 2018b)". We checked and did not find any repeated citations. |
| R1C14 | Line 44: the reference Wu et al. 2019 is missing in the list of references at the end. | The reference to Wu et al. 2019 was listed on lines 446–448 of the original manuscript. Sorry, we noted that it is not in an alphabetical order and we have now put it after the reference of Weedeon et al. (2014). |
| R1C15 | Lines 49-50: I would suggest omitting the sentence "However, it is noted that the model performance for the simulated bioenergy crop yield was not validated at all" as an argument to justify the novelty of the work. I doubt the authors are claiming that the previous two studies using H08 with representation of bioenergy yield ignored properly validating the model and this study contributes this novelty. I would suggest that authors replace this sentence with an explanation of how the new work builds on the two- step approach documented in the two previous papers (Yamagata et al. 2018 and Wu et al. 2019). | We agree with the Reviewer's concern and suggestion. The novelty of this paper lies in its systematic parameter calibration using the best available multi-site data. The first bioenergy crop implementation in H08 (Yamagata et al., 2018). Using the same bioenergy crop scheme, another recent study also used H08 estimates of yield for *Miscanthus* and switchgrass to predict global bioenergy potential (Wu et al., 2019). Our paper is based on the work of Yamagata et al. (2018). We have rephrased the sentence as follows (lines 47–48):

"Based on the work of Yamagata et al. (2018), here we improved the bioenergy crop simulation in H08 by performing a systematic parameter calibration for both *Miscanthus* and switchgrass using the best available data." |
| R1C16 | Line 61: 'The six sub-modules', You have not introduced what those six submodules are yet. I would start by listing them or at least list them in () right after this phrase. | Thank you. We have added the six submodules to the sentence, as follows (lines 62–64): |

| | | "The six sub-modules (land surface hydrology, river routing, crop growth, reservoir operation, environmental flow requirements, and anthropogenic water withdrawal) are coupled in a unique way (Fig. 1a)." |
|---|---|---|
| R1C17 | Line 75: I would expand on this section to show the two-step approach here before talking about model enhancements in the next section (2.2). Even if those were presented in the two previous publications, I would at least include them in SI to make this manuscript a standalone piece. | We have rephrased the enhancement section and included the two-step approach as follows (lines 125–133):

"The original bioenergy crop implementation in H08 (Yamagata et al., 2018) was conducted in two steps. First, crop parameters (see the old values in Table 2) for *Miscanthus* (refer to *Miscanthus giganteus* in this study) and switchgrass (refer to *Panicum virgatum* in this study) were adopted based on the settings from the SWAT model 2012 version (Arnold et al., 2013). However, the default parameters did not reflect the characteristics for *Miscanthus* and switchgrass well, which could lead to serious bias based on the result in Trybula et al. (2015). Second, maturity was defined by either undergoing an autumn freeze (i.e., the air temperature was below the minimum temperature for growth) or the exceedance of the maximum of 300 continuous days of growth. Because both *Miscanthus* and switchgrass are perennial, the potential heat unit was set as unlimited (see the old values in Table 2). However, this unlimited potential heat unit is far from the observations (see the new values in Table 2) reported by Trybula et al. (2015)." |
| R1C18 | Lines 76-85: I would suggest including all the equations and steps for how yield is simulated to shed more light about the method and to allow reproducibility of the approach. | We have added the equations and text related to yield simulation. Since this addition is quite long, we have not included it here; please see details in Section 2.1. |
| R1C19 | Line 90: 'as an output item' Are you saying that can you simulate water consumption as a new output variable? It is not clear. | You are correct. We have added a new output variable for water consumption to the crop module. |
| R1C20 | Line 91: 'Fifth, we fixed the bug in the original code'. What Bug? One could say 'we fixed a bug in the original code'. But this is | The bug is related to the improper use of ".eq." in place of ".ge." Since this is too |

| | | |
|---|---|---|
| | so vague and does not really give the reader any additional information. I would suggest dropping the fifth point. Such details are best documented in SI. | trivial to report, we have taken your suggestion removed it from the main text. |
| R1C21 | Lines 105-110: can you mention the number of data points and years being used? | We have added the numbers and years on lines 169–175, as follows:

"To independently calibrate and validate the performance of H08 in simulating the bioenergy yield, we collected and compiled up-to-date site-specific (varied from 1986 to 2011) and country-specific (varied from 1960 to 2010) yield data from both observations and simulations (Clifton-Brown et al., 2004; Searle and Malins, 2014; Heck et al., 2016; Kang et al., 2014; Li et al., 2018a). For *Miscanthus*, the yield data used covered 72 sites (64 rainfed and 8 irrigated; observed) and 15 countries (simulated). The simulated country-specific data is from MISCANMOD and LPJml. For switchgrass, the yield data used covered 57 sites (55 rainfed and 2 irrigated; observed) and 16 countries (simulated). The simulated country-specific data are from HPC-EPIC and LPJml." |
| R1C22 | Line 115: what variable is being calibrated here? H08 simulates many output variables. How does the calibration process ensure that the adopted calibration process does not offer a gain in better matching one variable at the expense of another variable? For example, did the authors calibrate runoff first and then yield, or is it done all at once? If it is the latter, then showing some results on runoff would be necessary. I am not asking the authors to necessarily do additional work, but rather to better explain their approach. | Here, we calibrated the five key parameters of radiation use efficiency (be), maximum leaf area index (blai), base temperature (Tb), maximum daily accumulation of temperature (Hunmax), and minimum temperature for planting (TSAW) that influence the yield simulation in the crop module. The standard H08 model uses a priori parameters; therefore we did not calibrate other variables such as runoff in the land surface hydrology module. |
| R1C23 | Line 117: 'the enhanced h08'. Does this mean that the second simulation was only done for the enhanced model? | You are correct. This simulation is used to analyze the effect of irrigation on yield, water consumption, and water use efficiency. Based on your suggestion (R1C34), we have reorganized the |

| | | | simulation setting section as follows (lines 187–193): |
|---|---|---|---|
| | | | "After calibration, four different kinds of simulation were run with different purposes. The first simulation was conducted using the original model without irrigation to investigate its performance. The second simulation was conducted using the enhanced model without irrigation to investigate its performance under rainfed condition. The third simulation was conducted using the enhanced model with irrigation to investigate its performance under irrigated condition. These three simulations were conducted at a daily scale with annual meteorological data from WFDEI for the period 1979–2016. The last simulation was conducted using identical model settings to the third one, except using different meteorological data from S14FD for the period 1979–2013. Note that irrigation in this study means uniform unconstrained irrigation." |
| R1C24 | | Lines 124-125: A bit unclear. Was the calibration done as a multi-objective optimization process to optimize both the RMSE and R values. For example, how do you decide an optimal parameter set when the two goodness-of-fit variables disagree? Figure 3 only shows RMSE, so I would suggest that you stick to this one and drop the R coefficient. Also, it is not clear if observed data is available for several individual years or only a single average year is available. If a time series exists, then I would suggest using goodness-of-fit measures such as Nash-Sutcliffe. | We apologize for the unclear description. Let us further explain the method. From a statistics perspective, root mean square error (RMSE) measures the standard deviation of prediction errors compared to the observations. The correlation coefficient (R) measures the correlation between the prediction and observation. Here, we gave the priority to RMSE, as it is a better metric for measuring errors in the predicted yield compared to R. We have added a figure showing the variations of RMSE and the corresponding R values in Fig. S1. It shows good agreement between the lowest RMSE and corresponding relatively high R.

The majority of the yield data fall within a single period instead of an individual year. Thank you for the suggestion of using the Nash-Sutcliffe model efficiency |

| | | coefficient, we did not use it due to the lack of time-series yield data. |
|---|---|---|
| R1C25 | Line 137: 'because the few sites that were irrigated'. Please rephrase. | We have modified the sentence, as follows (line 216):

"because only a few sites were irrigated" |
| R1C26 | Line 139: 'previous reports' Please add citations to support this claim. The single sentence that comes afterward is insufficient. What about other parameters? | Thank you for noting this issue. We have rephrased the sentence by adding a citation, as follows (line 217):

"These values are similar to those of Trybula et al. (2015)"

Other variables, such as base temperature, and maximum leaf area indices are also similar to those of Trybula et al. (2015). |
| R1C27 | Lines 140-145: how does this work differ from Trybula et al 2015? This is not discussed in the intro. Also given that the adopted approach follows the SWAT implementation in Trybula et al 2015, and almost all of the parameters taken from the literature are also taken from Trybula et al 2015, would not it guarantee that you get similar parameter values for the other calibrated values to match those in Trybula et al 2015? What about other studies? | We apologize for this, let us further explain it here. Basically, we conducted a global calibration and evaluation with the best available data, while the work of Trybula et al. (2015) is based on one site observation and validation. The work of Trybula et al. (2015) is the first report of updating the SWAT for bioenergy crop simulation based on field observations. It provides a valuable reference for our study, as the crop module of H08 is similar to that of SWAT. Therefore, in our model enhancement process, the crop parameters related to leaf area development (potential heat unit, optimum temperature, maximum leaf area index, and two complex number; see details in Table 1) were based on their field observations (Trybula et al., 2015).

For other parameters, including radiation use efficiency (be), maximum leaf area index (blai), base temperature (Tb), maximum daily accumulation of temperature (Hunmax), and minimum temperature for planting (TSAW), we conducted a systematic multi-site calibration and evaluation based on the parameter ranges reported in other studies |

| | | | (see Table 3). Our finalized parameters obtained through this approach are generally similar to those reported in Trybula et al. (2015), and are well within the range of other studies, as shown in Table 3. |
| --- | --- | --- | --- |
| R1C28 | Lines 148-154: although the results are better than the original version, the results still seem to show a tendency to underestimate based on the results shown in figure 3. | | Site-specific yield simulation and validation of traditional crops is a major challenge for global models (Müller et al., 2017), notwithstanding the bioenergy crop, which are being added to existing global models. For example, underestimation or overestimation have been reported in other global models like LPJml and ORCHIDEE that including the bioenergy crops. We added a new figure (Fig.3) of the calibrated results. It illustrates very good performance. Fig. 4 shows the validation of the model. Although it shows much better performance than the original simulation, it also shows a tendency toward underestimation. However, if we separately analyze each site, as shown in Fig. 5, most yield estimates were similar to or within the observed yield ranges. Therefore, our simulation appears to be reasonable at the global scale. We have further quantified the bias to illustrate the improvement of the model and rephrased the text as follows (lines 229–238):

"Points in a scatterplot comparing simulated yields derived from the enhanced H08 with observed yields are well distributed along the 1:1 line. It can be seen that the performance of the enhanced H08 was improved over that of the original H08. For *Miscanthus*, the bias of original model ranged from –84% to 80% with a mean of –52%, while the bias of the enhanced model ranged from –59% to 53% with a mean of –9%. For switchgrass, the bias for original model ranged from –78% to 338% with a mean of 25%, while the bias for the enhanced model ranged from –52% to 109% with a mean of – |

| | | 7%. Note that it also shows a tendency toward underestimation for some sites, especially for *Miscanthus*. More detailed site-specific results are shown in Figs. 5a (*Miscanthus*) and Fig. 5b (switchgrass). To depict the uncertainties in the observed yield, the minimum and maximum observed yields are shown as error bars in Fig. 5. It was found that the simulated yields were within or close to the range of the observed yields. The simulated relative error was randomly distributed, was substantially smaller than the range of the observed yields, and showed no climatic bias." |
|---|---|---|
| R1C29 | Line 158: 'well at sites 1, 2, and 10' so how many sites are under irrigation? You should mention it here. | There are ten sites with irrigation. We have modified the sentence, as follows (lines 240–241):

"We also investigated the performance under the irrigated condition (shown in Fig. 6). We used the reported observed yields for ten sites globally (Table S3)." |
| R1C30 | Lines 166-175: Did you drop the missing value from the significance test analysis (e.g., Finland in Fig 5d, Mongolia in Fig 5e)? I am still unsure whether the yield values from the other studies are average values over a particular period, and if it is the same period as in this study. | First, we did not drop the missing values. Note that the yield from MISCANMOD is reported with yield less than 10 Mg ha$^{-1}$ yr$^{-1}$ excluded (Clifton-Brown et al., 2004); therefore, we used the same method to make the comparison consistent. As the simulated yield for Finland is less than 10 Mg ha$^{-1}$ yr$^{-1}$, therefore there are no values for Finland. For Mongolia, our estimated value was 0.4 Mg ha$^{-1}$ yr$^{-1}$ and was rounded to 0 Mg ha$^{-1}$ yr$^{-1}$.

Second, based on your comment below (R1C32), we moved the text related to the study period, as follows (on lines 253–256):

"The periods of climate data used as inputs were 1960–1990, 1980–2010, and 1982–2005 for MISCANMOD, HPC-EPIC, and LPJmL, respectively. Here, the comparisons were conducted using exactly the same period as that of HPC-EPIC and LPJmL. For MISCANMOD, |

| | | however, we used the data from 1979–1990 due to data availability." |
|---|---|---|
| R1C31 | Lines 188-189: 'This can also be inferred from the validation results in Heck et al. (2016)' Please elaborate. | We have added an explanation, as follows (lines 276–277):

"This can also be inferred from the validation results (Fig. 1a) in Heck et al. (2016) since the LPJml-simulated yield is close to the yield of *Miscanthus* compared to those of switchgrass." |
| R1C32 | Lines 196-201: This information should appear earlier in the manuscript, so the reader is left wondering about such details. Also, if there is annual data from the other studies, then why not look at the timeseries instead of simply comparing the average value over a time period? To say a model can capture the long term mean over different basins is one level of validation, but to say that the model can also capture the interannual variability of yield from year to year, then this is a much more desirable level of validation. | As noted in a previous reply (R1C30), we have moved this text to the beginning of the section. Unfortunately, all values reported in previous studies are in mean annual terms. We used the average values for each component to ensure consistent comparison. |
| R1C33 | Lines 203-220: This section comes as s surprise as it was not mentioned earlier as part of the framing of the paper in the front sections. | We apologize. This section shows the spatial distribution of yield, which is helpful for clarifying its geographical differences among climate zones. Based on your suggestion, we have added a sentence (in bold below) to notify readers of this information in the last paragraph of the Introduction Section, as follows (lines 51–55):

"The following sections of this paper will: 1) describe the default biophysical process of the crop module in H08, 2) explain the enhancement of H08 for *Miscanthus* and switchgrass, 3) evaluate the enhanced performance of the model in simulating yields for *Miscanthus* and switchgrass, **4) map the spatial distributions of the yield of *Miscanthus* and switchgrass**, and 5) illustrate the effects of irrigation on the yield, water consumption, and WUE (defined here as the ratio of yield to water consumption) of *Miscanthus* and switchgrass." |

| | | |
|---|---|---|
| R1C34 | Line 206: It is confusing how many simulations were done in the study. The authors talk about two simulations twice, but are referring to different ones. I would suggest including an experimental design section as part of the methodology section to explain the different simulations to be conducted over a historical period (rainfed/irrigated, original/enhanced, ...). | Thank you for this constructive suggestion. We have reorganized the simulation setting in Table S1 and the description is now as follows (on lines 187–193):

"After calibration, four different kinds of simulation were run with different purposes. The first simulation was conducted using the original model without irrigation to investigate its performance. The second simulation was conducted using the enhanced model without irrigation to investigate its performance under rainfed condition. The third simulation was conducted using the enhanced model with irrigation to investigate its performance under irrigated condition. These three simulations were conducted at a daily scale with annual meteorological data from WFDEI for the period 1979–2016. The last simulation was conducted using identical model settings to the third one, except using different meteorological data from S14FD for the period 1979–2013. Note that irrigation in this study means uniform unconstrained irrigation." |
| R1C35 | Lines 211-220: The validation results shown and discussed in the main body of the manuscript only talk about the rainfed simulations. It is unconvincing to skip the validation step for the irrigation module, and then show results and draw conclusions using that irrigation modeling capability. In this section, results from this study are shown, but they are not contrasted with estimates from previous studies. | Following your comment (R1C5), we have added validation results under irrigated condition, as noted in a previous reply (R1C5). The results are comparable to previous reports, as the discussion on lines 307–310:

"The spatial distributions of yield increases due to irrigation simulated by H08 were very similar to those simulated by LPJmL (Beringer et al., 2011). At the continental scale (e.g., Europe), yield increases were located mainly in southern Europe, consistent with the findings obtained using MISCANMOD (Clifton-Brown et al., 2004)." |
| R1C36 | Lines 223-233: Is this based on some aggregated regions, or on all the grids that belong to each climate zone? How do these results compare to other studies that were discussed in section 3.4? assuming this was | This is a very good point. Yes, it is based on all grid cells belonging to specific climate zones. However, we used the results for grid cells with yield higher than 2 Mg ha$^{-1}$ yr$^{-1}$ (low-yield productivity). |

| | based on a grid-level analysis, why not plot the results for all the grids and show a scatter plot (yield on the y-axis, and aridity or some other index that allows for distinguishing among the different climate zones on the x-axis)? This would allow the authors to fit a line to the data and talk about the results in a more compelling way. | Based on your suggestion, we constructed a scatterplot diagram between yield and aridity (shown below). However, we found it difficult to directly differentiate the effect of climate. Meanwhile, our current figures clearly show the differences among different climate zones. This section provides additional analysis of the predicted yield, which may not affect the main conclusion of this study. Therefore, please let us retain the original presentation of these results.

 |
|---|---|---|
| R1C37 | Line 241: 'WUE, which is defined in this study as the ratio of yield to water consumption' This should have appeared the first time the term is mentioned in the manuscript. | We have defined this term in the introduction, as follows (lines 54–55):

"5) illustrate the effects of irrigation on the yield, water consumption, and WUE (defined here as the ratio of yield to water consumption) of *Miscanthus* and switchgrass." |
| R1C38 | Line 246: 'The WUE values for Miscanthus were higher than those for switchgrass, which is inconsistent with values in previous reports' Please add a sentence to articulate why? | We are sorry for using the incorrect word; it should be "consistent", as the results are similar (WUE of *Miscanthus* is higher than that of switchgrass). We have therefore changed the word "inconsistent" to "consistent". |
| R1C39 | Line 263: 'which was useful for optimizing bioenergy land with better consideration of water protection' – I am not sure what this means? | We have changed the sentence, as follows (lines 361–362):

"which was useful for bioenergy land-scenario design. For example, more land can be allocated to the areas with greater WUE." |
| R1C40 | Lines 266: 'and our results are reproducible with the transparent parameter disclosed.' | Thank you for noting this issue. We have deleted the sentence. |

| | | |
|---|---|---|
| | Just sharing the parameters sets does not guarantee reproducibility. I would suggest omitting that phrase. | |
| R1C41 | Line 277: why was not this yield map used in the previous sections as part of the validation exercise? Also, I would suggest moving figure S7 out of SI and into the main text. | We have added this yield map to the Method Section 2.4, as follows (lines 182–184):

"A global yield map of *Miscanthus* and switchgrass that was generated using a random-forest algorithm (Li et al., 2020) was also used to compare the results. This yield map provides a benchmark for evaluating model performance because it is largely constrained by the observed yield ranges, denoting the yields achievable under current technologies (Li et al., 2020)."

We also moved the corresponding result into the main text to Result Section 3.4, as follows:

"As shown in Fig. 8, we compared our simulation with the latest available global bioenergy crop yield map, generated from observations using a random-forest (RF) algorithm (Li et al., 2020). This RF yield map provides a benchmark for evaluating model performance because it is largely constrained by the observed yield ranges, denoting the yields achievable under current technologies (Li et al., 2020). As shown in Fig. 8a and Fig. 8b, there were small differences between our estimated yield and RF yield for switchgrass, whereas larger differences were found for *Miscanthus*, especially in tropical regions. There is a similar case for ORCHIDEE, as shown in Fig. S21 in Li et al. (2020). We also compared the differences in the mean values for *Miscanthus* and switchgrass because they are not distinguished in LPJmL. As shown in Fig. 8c and Fig. 8d, the differences between our estimations and the RF yields were generally lower than those between the LPJml estimations and RF yields. In summary, our |

| | | estimations were well within the ranges of those of ORCHIDEE and LPJml." |
|---|---|---|
| R1C42 | Figure 4: To be consistent with the black error bars, the blue/red ones should also reflect max/min. Also, why include all the years for observations? Should not these be for the years for which there is an associated observed yield value? | We have included the maximum and minimum values for *Miscanthus* (red) and switchgrass (blue) in the revised manuscript. Since the observed yields are from varying periods, we followed the methods of Heck et al. (2016), Beringer et al. (2011), and Li et al. (2018), comparing the mean simulated yield within a historical period to the observed yield. This was done in part due to missing records of harvest year for some observations. |
| R1C43 | Figure 7: why is the y-axis for panel b flipped around as if the values should be negative? I would suggest keeping it consistent with the other two panels (0 at the bottom left corner, and the bar chart goes upward for positive values). | We agree with your suggestion and have modified the y-axis in Fig. 7b. |
| R1C44 | Table 1: Please add another column to define the different parameters and what they mean physically. | Thank you. We have added a new table to show the definition and physical meaning. For details, please see Table 1. |

References:

[revised manuscript text omitted]

**Response to Reviewer 2**

| No. | Comment | Response |
|-----|---------|----------|
| R2C1 | This manuscript enhanced the capability of a global hydrological model named H08in simulating two perennial bioenergy crops, Miscanthus and switchgrass. The results were validated against site-level and country-level observed crop yields. The enhanced model is applied to simulate the impact of irrigation on crop water consumption and water use efficiency compared to rainfed condition. This study makes contribution to study the impact of large-scale deployment of bioenergy crops on water resources. However, I have some major comments as listed below. | Deer Reviewer, thank you very much for taking time to carefully read our manuscript. We are pleased to see your agreement on the contributions of this paper. Your valuable comments enabled us to clarify a number of points that we previously unaware of, and we hope that we have increased the quality of the manuscript substantially. We have revised the paper by trying to incorporate all relevant comments and remarks. We have also tried to respond to all the comments meticulously as you may see below. Please find our responses to each comment below. |
| R2C2 | Model validation: This study only validates the simulated yield results against observations for Miscanthus and switchgrass. While the main contribution/innovation of this study is on hydrological applications, this study didn't validate any variables for the water cycle, including evapotranspiration, runoff, and irrigation. With-out such validations, I feel difficult to be convinced for the reliability of the simulated results for crop water consumption and WUE. | Thank you for noting this issue. As you mentioned, we validated the simulated yield because our primary goal in this study was to improve the simulation of bioenergy crop yield in the H08 global hydrological model. Note that variables related to the water cycle, such as river discharge, terrestrial water storage, and water withdrawal have been thoroughly validated in a series of previous studies (Hanasaki et al., 2008a, 2008b, 2018). Here, we noted this has not been explicitly described in the manuscript, we therefore added it on lines 60–62. To address this question as well as possible, we compared our simulation of irrigation water consumption/withdrawal (on-going study) with previous reports (as shown in the table below), and found that our simulation is well within the range of existing reports. Because WUE is calculated using yield and water consumption, we believe that our estimates of WUE is also reasonable. |

| Studies | Irrigation water consumption/withdrawal [km$^3$ yr$^{-1}$] | |
|---|---|---|
| Beringer et al. (2011) | 1481~3880** | |
| Bonsch et al. (2016) | 3350*** | |
| Yamagata et al. (2018) | 1910** | |
| Heck et al. (2018) | > 2334** | |
| Jan et al. (2018) | 3000~9000** | |
| Stenzel et al. (2019) | 587~2946** | |
| Our study | 2187**/3929*** | |
| Note: ** refers to water consumption; *** refers to water withdrawal | | |

| | | | |
|---|---|---|---|
| R2C3 | Study innovation: The Introduction didn't well motivate the study and present the novelty/uniqueness of this study. For example, the argument "However, it is noted that the model performance for the simulated bioenergy crop yield was not validated at all." is a little bit difficult to be taken as an innovation of this study. And almost all the parameter values were directly taken from Trybula et al. 2015, which makes me wonder what are the main differences/improvements of this current study compared to Trybula et al. 2015? Given the difference between H08 used in this study and SWAT used in Trybula et al. 2015, can the authors justify the applicability of directly using SWAT's parameter values? | This is a good point. Let us further explain the uniqueness of our study here. Currently, only few models, such as LPJml, H08, and CLM5 include global implementations of both bioenergy and schemes for irrigation, river routing or water withdrawal. This limitation severely restricts the application of models to address possible global bioenergy–water tradeoffs or synergies in the future. Moreover, these three models have some limitations. First, in LPJml, *Miscanthus* and switchgrass are not distinguished and instead a general C4 grass is used to parameterize both species. Separate parametrization of these two bioenergy crops could enhance the bioenergy simulation, as they show major differences in plant characteristics and crop yield. Second, CLM5 has been successfully modified and validated for separate simulation of *Miscanthus* and switchgrass based on observations at the University of Illinois Energy Farm (Cheng et al., 2020), but global validation or application remain untested. Third, H08 has two weakness: 1) the original model produces apparent overestimations or underestimations, and 2) | |

| | | | |
|---|---|---|---|
| | | the original assumptions of potential heat units are unrealistic. Our study addressed these gaps and issues through systematic parameter calibration using the best available data. We have rephrased the introduction and provided further details on lines 30–50.

About the second question, the work of Trybula et al. (2015) is the first report of updating the SWAT for bioenergy crop simulation based on field observations. It provides a valuable reference for our study, as the crop module in H08 is similar to that of SWAT. Therefore, in our model enhancement process, the crop parameters related to leaf area development (potential heat unit, optimum temperature, maximum leaf area index, and two complex numbers; see details in Table 1) were based on their field observations (Trybula et al., 2015). For other important parameters, such as radiation use efficiency (be), maximum leaf area index (blai), base temperature (Tb), maximum daily accumulation of temperature (Hunmax), and minimum temperature for planting (TSAW), we conducted systematic calibration based on the ranges reported for that parameter in previous studies (see Table 3). The results demonstrated that the finalized parameter scheme is applicable to global simulation of bioenergy yield. It is possible to use SWAT's parameter because the crop module structure of H08 is similar to that of SWAT. | |
| R2C4 | Model description: This study only describes the crop module in H08 without much descriptions for the hydrological module in the model, especially given the important role of hydrological processes in this study. In addition, many indices and simulations (e.g., using new meteorological dataset) were not well described in the methods | Based on your suggestion, we have largely revised the methods section, as follows. First, we added the model structure to Fig. 1. The relevant hydrological processes are described on lines 62–69:

"The six sub-modules (land surface hydrology, river routing, crop growth, reservoir operation, | |

| | | |
|---|---|---|
| | section, such as how WUE is calculated, how irrigation works, and how many simulations were conducted in total and their respective purposes. | environmental flow requirements, and anthropogenic water withdrawal) are coupled in a unique way (Fig. 1a). The land surface module can simulate the main water cycle components, such as evapotranspiration and runoff. The former is used in the crop module, and the latter is used in the river routing and environmental flow modules. The agricultural water demand simulated by the crop module and the streamflow simulated by the river routing and reservoir operation modules finally enter into the withdrawal module. Note that the crop module is independent, except for the water stress calculations, which require evapotranspiration and potential evapotranspiration inputs from the land surface hydrology module." |
| | | Second, we added a description of the additional S14FD meteorological data on lines 164–166. |
| | | "Another meteorological dataset for the period 1979–2013 in S14FD (Iizumi et al. 2017) with the same spatial resolution was also used to check the stability of results to input meteorological data." |
| | | Third, we added the equations used for yield calculation on lines 79–123. Fourth, we described the calculation of WUE on lines 195–201. Fifth, we added a description on irrigation in lines 77–78. Sixth, we modified the simulation setting descriptions on lines 187–193. Sine these additions are quite long, we have not included them here; please see details in the specific lines noted above. |
| R2C5 | Paper organization: The main context is missing many important information (e.g., sensitivity test results, model descriptions, equations). Many important information and results were given in the SI rather than directly presented in the main context. The | Thank you. We have reorganized the paper, as follows. First, we added a schematic figure to show the submodules of H08 as Fig. 1a. The corresponding text is on lines 62–69. Second, we added the equations used for yield calculation of the crop |

| | | | |
|---|---|---|---|
| | methods section is missing descriptions for the simulations conducted in this study and many new simulations came out suddenly in the Results sections. It will be necessary to reorganize the paper and move some important information from SI to the main context. | | module on lines 79–123. Third, we described the sensitivity analysis in Section 2.7 and the result are presented in Table S5. Fourth, we rephrased the simulation setting description in Section 2.5 and added a summary table (Table S1). Fifth, we added meteorological data (S14FD) in Section 2.3. Sixth, we moved the original Fig. S5 and Fig. S7 to the main text (see Fig. 5 and Fig. 7, respectively, in the revised manuscript) and added corresponding text to Section 3.2 and 3.4. |
| R2C6 | Limitation in discussion: the current results and discussions are quite limited. For example, quantitative evaluations for model improvements were missing. What are the improvements of the enhanced H08 compared to its old version which uses C4 grass to characterize switchgrass and Miscanthus? One of the most important features of switchgrass and Miscanthus is their perennial features and longer growing seasons, but this study didn't have any discussions on this kind of perspectives. | | Thank you for this very good suggestion. The enhanced model strongly reduced the yield bias for both *Miscanthus* and switchgrass. Also, as noted in previous reports (Cheng et al., 2020), *Miscanthus* and switchgrass have longer growing seasons than maize. Here, we compared our results with reported growing season days. We added a discussion of these differences, as follows (lines 354–357): "Compared with the original H08, our enhanced model markedly decreased the mean bias (from –52% to –9% for *Miscanthus*, from 25% to –7% for switchgrass). Moreover, the growing seasons for *Miscanthus* (145–165) and switchgrass (101–114) during the period 2009–2011 at the Water Quality Field Station of the Purdue University Agronomy Center are consistent with the values of 140 and 120 reported in Trybula et al. (2015)." |
| R2C7 | Lines 31-36: Actually, CLM5 also has the irrigation scheme and river routing and CLM5 also includes both bioenergy crops and the water cycle. | | We have added CLM5, as follows (lines 30–32): "However, among these models, only a few, such as LPJml, H08, and CLM5 include the global implementation of schemes for irrigation, river routing or water withdrawal." |
| R2C8 | Line 34: typo, should be "bioenergy and the water cycle" | | Thank you. We have corrected the typo. |

| | | |
|---|---|---|
| R2C9 | Line 61: I am curious does it mean H08 can only simulate hydrological processes and crop growth as a 0.5 degree and at a daily scale? How about other spatial and temporal resolutions? Can H08 simulate GPP and LAI? If so, how about the simulation results for GPP and LAI? | You are correct. For global standard simulation (default setting), hydrological processes and crop growth can presently be simulated only at the 0.5-degree and daily. Regional versions have higher spatial resolution (5 arc-minutes) (Hanasaki et al., 2020). As the land surface hydrological model of H08 is the first generation that based on the bucket model (Manabe et al., 1969), GPP and LAI are not estimated in the land surface model. This is different from the LPJmL and CLM5, which are a dynamic vegetation model and a latest generation land surface vegetation model, respectively, and they do simulate GPP and LAI. In the crop module of H08, it calculates the yield and LAI. LAI is coded as a medium variable in the process of yield calculation but is not an output item in current model version. Since our primary goal here is the improvement and validation of bioenergy crop yield, please let us retain current model version. We will consider your comment and modify the code in future model development. |
| R2C10 | Lines 61-64: What are the six sub-modules? It will be great if the authors can add more descriptions for the H08 model (e.g., calculations/illustrations for the hydraulic processes), as not every reader is familiar with H08. | We have added a description of the six submodules after the term on lines 60-62. We have also added a schematic diagram showing the connections among submodules as Fig. 1a. We have included the equations related to the yield simulation in the crop module on lines 80–120. A full description of the H08 model would require thousands of words, and is available elsewhere (Hanasaki et al., 2008a, 2008b, 2018). |
| R2C11 | Line 75: what is single-irrigated and double-irrigated mean? | Single-irrigated indicates that the irrigated land is used only for one crop per year, while double-irrigated refers to irrigated land planted with two crops per year. |
| R2C12 | Line 85: what is "substantially" mean? Can you quantify the changes? | Here, substantially represents a large difference between the modified (1830 °C for *Miscanthus*, and 1400 °C for switchgrass) and original (9999 °C for both |

| | | *Miscanthus* and switchgrass) potential heat units. |
|---|---|---|
| R2C13 | Section 2.2: Can you change some descriptions into equations? For example, how did you calculate the output item for water consumption and WUE? If they are already in the supplementary materials, it will be great if you can move some key equations to the main context. What is the bug in the original code? | We added the equations related to the yield calculation on lines 79–123. Water consumption is calculated as actual evapotranspiration. The bug is related to the improper use of ".eq." in place of ".ge." As this bug is too trivial to report, we have removed it from text. |
| R2C14 | Section 2.5: How is irrigation calculated in H08, such as the irrigated area and irrigated amount? | Our intention was to determine the general effects of irrigation on bioenergy crop yield and the variations among different climate zones. Therefore, we assumed a whole grid is irrigated for bioenergy crop production. The irrigation water amount in H08 is defined as the supply of water other than precipitation to maintain soil moisture above 75% of field capacity during the cropping period. |
| R2C15 | Line 23 under section 2.5: since 1944 simulations were conducted, can you give more results for the ensemble runs rather than just present the one with lowest RMSE? For example, what are the uncertainty ranges for the calibrations? What are the sensitivity results for all the calibrated parameters? Here the authors only mentioned the most sensitive parameter names in line 20 but no results were given to support it. | We have added a new figure (Fig. 3) to illustrate the performance of the enhanced model after calibration, which shows good agreement with the observations. We have also added a new figure (Fig. S1) showing the variations of root mean square error (RMSE) and corresponding correlation coefficient (R) values used for the calibration. The uncertainty range of each parameter is listed in Table 3. We also calculated the sensitivity and summarized the results in Table S5. Among the five parameters we calibrated, radiation use efficiency was the most sensitive parameter to the results, followed by base temperature. This finding is consistent with the sensitivity results reported by Trybula et al. (2015). |
| R2C16 | Line 38 in section 3.1: change to "because only few sites were irrigated". | We have changed the text, as follows (line 216):

"because only a few sites were irrigated" |

| | | |
|---|---|---|
| R2C17 | Line 38 in section 3.1: can you add reference after the "previous reports"? | We have rephrased the sentence and added a citation, as follows (line 217):

"These values are similar to those of Trybula et al. (2015)." |
| R2C18 | Line 50 in section 3.2: can the authors add more quantitative results and discuss the reasons/mechanisms for why the over- and under-estimations have been addressed in the enhanced H08? Actually Miscanthus is still underestimated in the enhanced H08, why? | We have quantified the bias and rephrased the text as follows (lines 231–233):

"For *Miscanthus*, the bias of original model ranged from –84% to 80% with a mean of –52%, while the bias of the enhanced model ranged from –59% to 53% with a mean of –9%. For switchgrass, the bias for original model ranged from –78% to 338% with a mean of 25%, while the bias for the enhanced model ranged from –52% to 109% with a mean of –7%."

One important reason for the improved bias is the adjustment of the potential heat unit based on the field observations from Trybula et al. (2015). This parameter adjustment would change the crop leaf area development and also the aboveground biomass accumulation. As for switchgrass, another important reason is the decrease of radiation use efficiency, which can largely address the overestimation.

Note that site-specific yield simulation and validation of traditional crops is a major challenge for global models (Müller et al., 2017), notwithstanding the bioenergy crop, which are being added to existing global models. For example, underestimation or overestimation have also been reported in other global models like LPJml and ORCHIDEE that including the bioenergy crops. We added a new figure (Fig.3) of the calibrated results. It illustrates very good performance. Fig. 4 shows the validation of the model. Although it shows much better performance than the original simulation, it also shows a tendency toward underestimation. However, if we separately analyze each site, as shown in Fig. 5, most |

| | | | |
|---|---|---|---|
| | | | yield estimates were similar to or within the observed yield ranges. Therefore, our simulation appears to be reasonable at the global scale. |
| R2C19 | Line 58 in section 3.2: what are sites 1, 2, and 10? Can you refer to more specific names or descriptions for those sites, as these site numbers are not quite meaningful? | | We have modified the text, as follows (lines 240–242):

"We also investigated the performance under the irrigated condition (shown in Fig. 6). We used the reported observed yields for ten sites globally (Table S3). We found that the simulated yields were within or close to the observed yields for five sites located in China, the UK, and France (see Table S3)" |
| R2C20 | Line 59 in section 3.2: again, adding irrigation scheme in H08 in the methods section will be helpful. | | We have added a description of the irrigation scheme on lines 77–78:

"Irrigation in H08 is defined as the supply of water other than precipitation to maintain soil moisture above 75% of field capacity during the cropping period." |
| R2C21 | Line 64 in section 3.2: the two results were similar, but what are the implications? What are the differences between the two meteorological datasets? Also, it makes me wonder how many simulations or how many kinds of simulations were conducted in this study? This new simulation with additional meteorological dataset never mentioned in the methods section. I will suggest the author add a new table or at least a new paragraph in the methods section to better illustrate the simulations conducted in this study, including their names, descriptions, differences, purposes, etc. | | We apologize for the unclear description. Let us further clarify the text here First, we aimed to test the stability of the modelling results by varying the meteorological inputs. The results indicated that our simulation is quite stable. The S14FD dataset is reported to be more accurate than WFDEI for representing the observed temperature and precipitation extremes in recent decades (1961–2000 and 1979–2008) (Iizumi et al., 2017). Second, four types of simulations were conducted, and we have added a new table (Table S1) and rephrased the text to describe the simulations as follows (lines 187–193):

"After calibration, four different kinds of simulation were run with different purposes. The first simulation was conducted using the original model without irrigation to investigate its performance. The second simulation was conducted using the enhanced model without irrigation to investigate its performance under |

| | | rainfed condition. The third simulation was conducted using the enhanced model with irrigation to investigate its performance under irrigated condition. These three simulations were conducted at a daily scale with annual meteorological data from WFDEI for the period 1979–2016. The last simulation was conducted using identical model settings to the third one, except using different meteorological data from S14FD for the period 1979–2013. Note that irrigation in this study means uniform unconstrained irrigation." |
|---|---|---|
| R2C22 | Section 3.3: can you add those correlation and significant level values in Figure 5 as well? | Thank you. We have added the corresponding correlations and significance values. |
| R2C23 | Line 10 in section 3.4: grammar error for the sentence | Thank you for letting us know about this issue. This section now reads as follows (line 307):

"indicating that irrigated yield was more than double the rainfed yield."

We have revised the whole manuscript further, and have employed the professional English proofreading service from Textcheck (http://www.textcheck.com/en/text/page/index). |
| R2C24 | Line 55-58 in section 3.6: again, how is the current results compared to old H08 which uses C4 grass to represent Miscanthus and switchgrass? | In response to the Reviewer's previous comment (R2C6), we have added the following discussions (lines 354–357):

"Compared with the original H08, our enhanced model markedly decreased the mean bias (from –52% to –9% for *Miscanthus*, from 25% to –7% for switchgrass). Moreover, the growing seasons for *Miscanthus* (145–165) and switchgrass (101–114) during the period 2009–2011 at the Water Quality Field Station of the Purdue University Agronomy Center are consistent with the values of 140 and 120 reported in Trybula et al. (2015)." |

| | | |
|---|---|---|
| R2C25 | Line 63-65 in section 3.6: I doubt the argument that the enhanced H08 is the only model that can simultaneously simulate Miscanthus and switchgrass with consideration of water management, as CLM5 also has this capability. | We apologize. We have added CLM5 to the introduction and modified this sentence, as follows (lines 362–364):

 "In summary, our enhanced model provides a new tool that can simultaneously simulate *Miscanthus* and switchgrass with consideration of water management" |
| R2C26 | Tables 1 and 2: could you add the long name or descriptions for these parameters? What is "step" mean in Table 2? | This is a good point. We have added a new table (Table 1) to describe the parameters, and their full names and descriptions can be found there. The term "step" refers to the increment of the parameter within the range of our calibration. We have changed the term "step" to "increment". |
| R2C27 | Figure 1: could you add a flow chart or schematic figure for the hydrological processes in H08 or the overall model structure? | We have added a schematic diagram to show the structure and connection of the submodules as Fig. 1b. |
| R2C28 | Figure 3: can the authors decrease the maximum magnitudes for figure b and d, like to be 40, since no data exceeds 40 and right now most of the points are centered to a very small range? And can the authors add a third axis (e.g., different colors) to distinguish the locations/climate zones for the points? | Thank you for this useful suggestion. We have modified the maximum value of the axis as you suggested. Since we had used red and blue colors to distinguish *Miscanthus* and switchgrass, we used different shapes (see the legend for Fig. 4) to identify the climate zone of each point. |
| R2C29 | Figure 6: it will be helpful to add a title name in the figure, e.g., (a) Rainfed Miscanthus. | This is a good point. We have added a title name in the upper right conner of the figure. For details, please see Fig. 9. |
| R2C30 | Figure 7: it may be helpful to move Figure S6 to the main context and combined with Figure 2 to better illustrate the methods section. But the authors can decide after revise the methods section. | Thank you for this useful suggestion. We have moved Fig. S6 to the main text and combined it with Fig. 2 to better illustrate the method. By doing this, we now include both climate zone and site location in Fig. 2. |

[revised manuscript text omitted]

5    Table S1. Different types of simulations in this study.

| ID | Water management | Model | Meteorological dataset | Purpose |
|---|---|---|---|---|
| 1 | Rainfed | Original model | WFDEI | To evaluate the performance of the original H08. |
| 2 | Rainfed | Enhanced model | WFDEI | To evaluate the performance of the enhanced H08 under rainfed condition. To calculate water use efficiency. |
| 3 | Irrigated | Enhanced model | WFDEI | To evaluate the performance of the enhanced H08 under irrigated condition. To calculate water use efficiency. |
| 4 | Rainfed | Enhanced model | S14FD | To investigate the variability of the result. |

| ID | Water management | Model | Meteorological dataset | Purpose |
|---|---|---|---|---|

Table S2+. Location and yield of the sites for *Miscanthus* (specified in Fig. 4) under rainfed condition.

| ID | Country | Longitude | Latitude | Minimum yield [t ha$^{-1}$ yr$^{-1}$] | Maximum yield [t ha$^{-1}$ yr$^{-1}$] | Mean yield [t ha$^{-1}$ yr$^{-1}$] | Reference |
|---|---|---|---|---|---|---|---|
| 1 | Indonesia | 107.70 | -7.00 | 31.9 | 31.9 | 31.9 | Blair et al. 1986 |
| 2 | US | -97.10 | 36.10 | 12.4 | 13.1 | 12.8 | Aravindhakshan et al. 2010 |
| 3 | US | -88.67 | 37.45 | 18.0 | 42.3 | 30.2 | Arundale et al., 2014a, 2014b; Heaton et al., 2008 |
| 4 | Turkey | 33.23 | 38.17 | 1.50 | 13.19 | 7.0 | Acaroğlu and Aksoy, 2005 |
| 5 | US | -88.39 | 38.38 | 19.00 | 47.00 | 31.4 | Arundale et al., 2014a, 2014b; Heaton et al., 2008 |
| 6 | US | -90.82 | 39.81 | 12.00 | 36.00 | 25.0 | Arundale et al., 2014a, 2014b; Heaton et al., 2008 |
| 7 | US | -88.23 | 40.08 | 22.0 | 45.5 | 33.8 | Arundale et al., 2014a; Heaton et al., 2008 |
| 8 | US | -88.19 | 40.17 | 17.00 | 24.10 | 20.6 | Wang et al., 2012 |
| 9 | US | -88.85 | 41.85 | 5.00 | 29.90 | 17.3 | Arundale et al., 2014a, 2014b; Heaton et al., 2008 |
| 10 | Italy | 10.32 | 43.67 | 9.00 | 48.00 | 26.2 | Angelini et al., 2009; Ercoli et al., 1999; o Di Nasso et al., 2011 |
| 11 | Switzerland | 9.13 | 47.57 | 14.00 | 14.00 | 14.0 | Poeplau and Don, 2014 |
| 12 | Germany | 10.00 | 48.00 | 20.00 | 20.00 | 20.0 | Lewandowski and Heinz, 2003 |
| 13 | Austria | 14.22 | 48.11 | 15.50 | 24.50 | 20.0 | Schwarz, 1993 |
| 14 | Germany | 9.97 | 48.13 | 12.70 | 16.50 | 15.0 | Lewandowski and Kicherer, 1997 |
| 15 | Austria | 14.15 | 48.14 | 13.20 | 24.40 | 18.8 | Schwarz, 1993 |
| 16 | Austria | 16.39 | 48.18 | 0.80 | 21.50 | 12.5 | Schwarz, 1993 |
| 17 | Austria | 15.55 | 48.19 | 2.00 | 23.84 | 14.9 | Schwarz, 1993; Schwarz et al., 1994a |
| 18 | Austria | 15.00 | 48.30 | 17.4 | 24.5 | 21.0 | Schwartz, 1993 |
| 19 | Germany | 11.54 | 48.31 | 0.41 | 20.88 | 12.6 | Schwarz et al., 1994b |
| 20 | Germany | 10.26 | 48.49 | 1.11 | 23.42 | 13.4 | Schwarz et al., 1994b |
| 21 | Germany | 11.63 | 48.60 | 0.28 | 20.43 | 10.2 | Schwarz et al., 1994b |
| 22 | Germany | 9.00 | 48.70 | 19.9 | 26.4 | 23.2 | Clifton-Brown et al., 2001a |
| 23 | Germany | 8.93 | 48.73 | 14.50 | 18.00 | 16.3 | Boehmel et al., 2008 |
| 24 | Germany | 8.92 | 48.75 | 5.60 | 30.50 | 13.7 | Gauder et al., 2012 |
| 25 | Germany | 9.19 | 48.78 | 0.51 | 22.54 | 11.2 | Schwarz et al., 1994b |
| 26 | Germany | 8.10 | 49.00 | 17.00 | 17.00 | 17.0 | Lewandowski el al., 2003 |
| 27 | Germany | 6.72 | 49.82 | 15.00 | 15.00 | 15.0 | Poeplau and Don, 2014 |
| 28 | France | 3.00 | 49.87 | 19.00 | 28.00 | 23.1 | Strullu et al., 2011 |
| 29 | France | 3.01 | 49.87 | 14.30 | 28.40 | 22.2 | Cadoux et al., 2014 |
| 30 | Germany | 9.90 | 49.90 | 6.2 | 19.8 | 13.0 | Kahle et al., 2001 |
| 31 | Germany | 10.77 | 50.97 | 15.00 | 15.00 | 15.0 | Poeplau and Don, 2014 |
| 32 | Blegium | 3.80 | 51.00 | 0.50 | 25.70 | 12.1 | Muylle et al., 2015 |
| 33 | UK | -1.26 | 51.10 | 0.80 | 23.50 | 14.5 | Price et al., 2004 |
| 34 | Poland | 22.63 | 51.23 | 0.44 | 29.43 | 13.1 | Borkowska and Molas, 2013 |
| 35 | Germany | 6.70 | 51.50 | 17.5 | 28.8 | 23.2 | Heaton et al. 2008 |
| 36 | Germany | 6.70 | 51.52 | 1.00 | 20.70 | 12.5 | Himken et al., 1997 |
| 37 | Germany | 7.62 | 51.78 | 1.47 | 18.44 | 10.0 | Schwarz et al., 1994b |
| 38 | UK | -0.40 | 51.80 | 9.8 | 17.8 | 13.8 | Christian et al. 2008 |
| 39 | UK | -2.64 | 51.80 | 13.00 | 24.00 | 18.0 | Price et al., 2004 |
| 40 | UK | -0.35 | 51.80 | 0.10 | 18.70 | 9.1 | Clifton-Brown et al., 2001a |
| 41 | UK | -0.36 | 51.82 | 12.00 | 14.50 | 12.9 | Richter et al., 2008 |

| 42 | UK | -0.62 | 52.01 | 13.70 | 16.20 | 15.0 | Richter et al., 2008 |
|----|----|-------|-------|-------|-------|------|----------------------|
| 43 | UK | -0.03 | 52.25 | 0.20 | 17.00 | 10.6 | Price et al., 2004 |
| 44 | Poland | 16.92 | 52.42 | 5.50 | 23.70 | 11.2 | Jezowski, 2008; Jezowski et al., 2011 |
| 45 | UK | 0.09 | 52.42 | 11.50 | 22.50 | 18.4 | Price et al., 2004 |
| 46 | UK | -4.02 | 52.43 | 0.30 | 17.20 | 10.6 | Zatta et al., 2014 |
| 47 | Germany | 10.80 | 52.60 | 8.8 | 13.5 | 11.2 | Kahle et al. 2001 |
| 48 | Germany | 8.26 | 52.61 | 2.10 | 20.02 | 10.1 | Schwarz et al., 1994b |
| 49 | Germany | 10.81 | 52.62 | 3.72 | 23.89 | 14.2 | Schwarz et al., 1994b |
| 50 | Ireland | -7.83 | 52.65 | 4.20 | 16.30 | 11.5 | Clifton-Brown et al., 2001b |
| 51 | Ireland | -7.27 | 52.67 | 2.00 | 15.80 | 9.4 | Clifton-Brown et al., 2001b |
| 52 | Germany | 8.81 | 52.68 | 3.46 | 19.01 | 9.6 | Schwarz et al., 1994b |
| 53 | Netherlands | 7.06 | 52.88 | 21.8 | 21.8 | 21.8 | van der Werf et al. 1993 |
| 54 | UK | -3.78 | 53.22 | 14.90 | 22.20 | 18.6 | Price et al., 2004 |
| 55 | Netherlands | 6.95 | 53.30 | 13.00 | 13.00 | 13.0 | Poeplau and Don, 2014 |
| 56 | Poland | 19.38 | 53.78 | 5.80 | 28.00 | 13.8 | Jezowski et al., 2011 |
| 57 | Germany | 12.60 | 53.90 | 7.5 | 12.6 | 10.1 | Kahle et al. 2001; Beuch et al., 2000 |
| 58 | UK | -1.11 | 54.12 | 0.50 | 13.00 | 7.8 | Price et al., 2004 |
| 59 | UK | -0.64 | 54.12 | 0.50 | 16.00 | 7.3 | Price et al., 2004 |
| 60 | Denmark | 9.12 | 54.90 | 6.20 | 14.00 | 10.0 | Schwarz et al., 1994b |
| 61 | Sweden | 14.00 | 56.00 | 0.10 | 24.70 | 6.6 | Clifton-Brown et al., 2001a |
| 62 | UK | -3.06 | 56.46 | 10.20 | 10.20 | 10.2 | Richter et al., 2008 |
| 63 | Denmark | 9.60 | 56.50 | 9.7 | 16.8 | 13.3 | Clifton-Brown et al. 2001a, 2004; Lewandowski el al., 2003 |
| 64 | Denmark | 9.40 | 56.80 | 7.7 | 8.9 | 8.3 | Jørgensen, 1997 |

Table S32. Location and yield of the sites for switchgrass (specified in Fig. 4) under rainfed condition.

| ID | Country | Longitude | Latitude | Minimum yield [t ha$^{-1}$ yr$^{-1}$] | Maximum yield [t ha$^{-1}$ yr$^{-1}$] | Mean yield [t ha$^{-1}$ yr$^{-1}$] | Reference |
|---|---|---|---|---|---|---|---|
| 1 | US | -97.70 | 28.45 | 4.50 | 13.00 | 7.8 | Muir et al., 2001 |
| 2 | US | -89.94 | 30.30 | 6.00 | 16.00 | 10.1 | Arundale et al., 2014a, 2014b; Heaton et al., 2008 |
| 3 | US | -87.32 | 32.00 | 1.52 | 12.07 | 5.9 | Bransby et al., 1990 |
| 4 | US | -98.20 | 32.23 | 1.50 | 21.50 | 9.9 | Muir et al., 2001; Sanderson et al., 1999 |
| 5 | US | -85.90 | 32.44 | 3.71 | 34.60 | 11.3 | Ma et al., 2001; Sladden et al., 1991 |
| 6 | US | -85.65 | 32.82 | 3.43 | 9.67 | 7.0 | Bransby et al., 1990 |
| 7 | US | -87.87 | 33.88 | 0.44 | 13.39 | 8.4 | Bransby et al., 1990 |
| 8 | US | -85.97 | 34.28 | 2.04 | 9.93 | 5.9 | Bransby et al., 1990 |
| 9 | US | -88.90 | 35.60 | 7.8 | 16.9 | 12.4 | Lemus, 2004 |
| 10 | US | -78.70 | 35.70 | 5.1 | 16.7 | 10.9 | Lemus, 2004 |
| 11 | US | -83.95 | 35.88 | 11.40 | 23.20 | 18.0 | Reynolds et al., 2000 |
| 12 | US | -84.00 | 35.90 | 11.2 | 24.9 | 18.1 | Lemus, 2004 |
| 13 | US | -97.07 | 36.12 | 8.31 | 13.82 | 11.0 | Aravindhakshan et al., 2011 |
| 14 | China | 109.32 | 36.85 | 2.36 | 16.55 | 8.6 | Xu et al., 2005, 2008 |
| 15 | US | -78.23 | 36.92 | 5.20 | 8.60 | 7.5 | Parrish et al., 1990 |
| 16 | US | -87.80 | 37.10 | 8.4 | 17.0 | 12.7 | Lemus, 2004 |
| 17 | US | -80.40 | 37.20 | 9.5 | 27.4 | 18.5 | Lemus, 2004 |
| 18 | US | -88.67 | 37.45 | 7.80 | 18.00 | 11.2 | Arundale et al., 2014a, 2014b; Heaton et al., 2008 |
| 19 | China | 118.49 | 37.46 | 3.46 | 4.51 | 3.8 | Gao et al., 2016: |
| 20 | US | -77.97 | 38.02 | 7.00 | 16.20 | 11.6 | Parrish et al., 1990: |
| 21 | US | -78.10 | 38.20 | 11.2 | 20.4 | 15.8 | Lemus, 2004 |
| 22 | US | -88.39 | 38.38 | 4.00 | 16.00 | 11.4 | Arundale et al., 2014a, 2014b; Heaton et al., 2008 |
| 23 | US | -88.96 | 38.95 | 4.00 | 15.00 | 9.7 | Arundale et al., 2014a; Heaton et al., 2008 |
| 24 | China | 113.18 | 39.55 | 4.40 | 9.30 | 6.9 | Xiong et al., 2008 |
| 25 | US | -79.90 | 39.60 | 12.8 | 20.5 | 16.7 | Lemus, 2004 |
| 26 | US | -90.82 | 39.81 | 8.00 | 15.00 | 10.3 | Arundale et al., 2014a, 2014b; Heaton et al., 2008 |
| 27 | US | -75.38 | 39.92 | 2.82 | 12.50 | 7.0 | Stout et al., 1988 |
| 28 | US | -96.77 | 39.99 | 1.90 | 15.69 | 6.2 | Sanderson et al., 1999 |
| 29 | US | -88.23 | 40.08 | 10.60 | 18.00 | 14.1 | Arundale et al., 2014a; Heaton et al., 2008 |
| 30 | China | 116.12 | 40.19 | 4.20 | 5.90 | 5.2 | Hou et al., 2010 |
| 31 | US | -78.00 | 40.70 | 3.3 | 9.4 | 6.4 | Sanderson, 2008 |
| 32 | US | -93.42 | 40.97 | 5.80 | 17.40 | 10.2 | Anderson et al., 1994 |
| 33 | US | -93.40 | 41.00 | 6.8 | 13.1 | 10.0 | Lemus et al., 2002 |
| 34 | US | -83.07 | 41.37 | 2.30 | 7.70 | 5.1 | Wright, 1990; Wright and Turhollow, 2010 |
| 35 | US | -83.05 | 41.50 | 3.00 | 9.00 | 5.4 | Wright, 1990; Wright and Turhollow, 2010 |
| 36 | US | -88.85 | 41.85 | 4.00 | 14.10 | 9.0 | Arundale et al., 2014a, 2014b; Heaton et al., 2008 |
| 37 | US | -88.90 | 41.90 | 10.4 | 12.5 | 11.5 | Heaton et al., 2008 |
| 38 | US | -100.00 | 42.00 | 5.0 | 7.4 | 6.2 | Schmer et al., 2010 |
| 39 | US | -93.77 | 42.02 | 4.90 | 15.90 | 9.6 | Anderson et al., 1994 |
| 40 | US | -76.45 | 42.45 | 0.89 | 13.11 | 6.7 | Pfeifer et al., 1990 |

| 41 | US | -77.00 | 42.87 | 1.17 | 7.60 | 4.4 | Pfeifer et al., 1990 |
|----|----|--------|-------|------|------|-----|---------------------|
| 42 | US | -99.80 | 43.70 | 0.8 | 5.9 | 3.4 | Mulkey et al., 2006 |
| 43 | US | -100.00 | 44.00 | 4.2 | 8.8 | 6.5 | Schmer et al., 2010 |
| 44 | US | -96.70 | 44.20 | 1.0 | 6.0 | 3.5 | Mulkey et al., 2006 |
| 45 | US | -100.00 | 44.28 | 5.00 | 5.00 | 5.0 | Hong et al., 2013 |
| 46 | US | -96.77 | 44.32 | 7.50 | 7.50 | 7.5 | Hong et al., 2013 |
| 47 | Italy | 11.50 | 44.40 | 7.9 | 11.5 | 9.7 | Di Virgilio et al., 2007 |
| 48 | US | -73.75 | 45.47 | 1.65 | 17.21 | 9.6 | Madakadze et al., 1998a, 1998b, 1998c; 1999 |
| 49 | US | -95.88 | 45.60 | 4.80 | 4.80 | 4.8 | Hong et al., 2013 |
| 50 | US | -97.23 | 46.65 | 3.50 | 12.80 | 9.4 | Meyer et al., 1994 |
| 51 | US | -97.02 | 46.95 | 7.30 | 10.30 | 9.0 | Meyer et al., 1994 |
| 52 | US | -100.00 | 47.00 | 5.6 | 5.8 | 5.7 | Schmer et al., 2010 |
| 53 | Germany | 8.93 | 48.73 | 8.00 | 14.00 | 11.3 | Meyer et al., 1994 |
| 54 | Blegium | 3.80 | 51.00 | 2.50 | 15.90 | 9.9 | Muylle et al., 2015 |
| 55 | UK | -0.35 | 51.80 | 1.19 | 13.97 | 6.8 | Christian et al., 2002 |

Table S43. Location and yield of the sites for *Miscanthus* and switchgrass (specified in Fig. S1) under irrigated condition.

| ID | Country | Longitude | Latitude | Minimum yield [t ha$^{-1}$ yr$^{-1}$] | Maximum yield [t ha$^{-1}$ yr$^{-1}$] | Mean yield [t ha$^{-1}$ yr$^{-1}$] | Reference |
|----|---------|-----------|----------|---------|---------|------|-----------|
| 1 | China | 108.06 | 34.27 | 3.5 | 44.2 | 16.7 | Ma et al., 2011 |
| 2 | China | 106.47 | 36.01 | 2.8 | 10.6 | 6.2 | Ma et al., 2011 |
| 3 | Italy | 14.35 | 37.38 | 1.2 | 30.6 | 18.0 | Mantineo et al., 2009 |
| 4 | Italy | 15.06 | 37.42 | 3.9 | 27.0 | 16.7 | Cosentino et al., 2007 |
| 5 | Turkey | 32.5 | 38.0 | 12.0 | 13.2 | 12.6 | Acarglu and Aksoy, 2005 |
| 6 | Portugal | -9.22 | 38.72 | 4.6 | 37.8 | 17.1 | Clifton-Brown et al., 2001a |
| 7 | Greece | 22.75 | 39.40 | 20.0 | 31.4 | 25.6 | Danalatos et al., 2007 |
| 8 | France | 3.00 | 49.88 | 4.8 | 32.5 | 7.7 | Zub et al., 2011 |
| 9 | UK | 0.40 | 51.70 | 8.3 | 19.7 | 14.0 | Beale and Long, 1995 |
| 10 | UK | 0.43 | 51.73 | 19.4 | 19.4 | 19.4 | Beale and Long, 1995 |

Table S5. Sensitivity analysis results of the parameters for (a) *Miscanthus* and (b) switchgrass

| Parameter | Sensitivity index | |
| --- | --- | --- |
| | *Miscanthus* | switchgrass |
| be | 0.60 | 1.27 |
| Tb | 0.40 | 0.66 |
| blai | 0.03 | 0.17 |
| Hunmax | 0.21 | 0.02 |
| TSAW | 0.07 | 0.00 |

| Parameter | Sensitivity index | |
| --- | --- | --- |
| | *Miscanthus* | switchgrass |
| be | 0.60 | 1.27 |
| Tb | 0.40 | 0.66 |

[Figure]

Fig. S1 Variation of the RMSE and corresponding R for the calibration.

[Figure]

Fig. S1 Site-specific performance (shown with increasing latitude from the bottom of the vertical axis) of the simulated yield (sim.) obtained using the enhanced H08 model compared with observed yields (obs.) for *Miscanthus* (mis.) and switchgrass (swc.) under irrigated condition. The longitude and latitude of each location ID for *Miscanthus* and switchgrass are given in Tables S3. Observation indicates the observed mean yield. The error bar (in black) represents the range of the observed minimum and maximum yield. The error bar (in red) represents the standard deviation of the simulated yield from 1979 to 2016.

[Figure]

Fig. S2 Box plots showing the first (lower line), median (solid line) and third (upper line) quartiles of the yield for observed (OBS.) and simulated (with meteorological data driven by WFDEI and S14FD) *Miscanthus* and Switchgrass. The mean value is indicated by the red line.

[Figure]

Fig. S3 Independent country-specific comparison of simulated yields from the enhanced H08 model and LPJmL under irrigated conditions.

[Figure]

50    Fig. S4 Country-specific comparison of the simulated yields of *Miscanthus* and Switchgrass from the enhanced H08 model with the ensemble yield of LPJmL under rainfed (a) and irrigated conditions (b), respectively.

[Figure]

Fig. S5 Spatial distribution of averaged annual precipitation (mm yr$^{-1}$) from 1979 to 2016.

[Figure]

55    Fig. S6 Five different kinds of Köppen climate zones based on the average meteorological data from 1979 to 2016. The specific categories are as follows: 1 (dark blue) for tropical climate zone; 2 (light blue) for dry climate zone; 3 (green) for temperature climate zone; 4 (yellow) for continental climate zone; 5 (red) for polar climate zone.

[Figure]

60 Fig. S7 Comparison of yield difference (simulated yield minus RF yields) between model simulations and the RF map (Li et al., 2020): a) for *Miscanthus* with the yield from H08 minus that from RF, b) for *Switchgrass* with the yield from H08 minus that from RF, c) for the mean of *Miscanthus* and Switchgrass with the yield from H08 minus that from RF, d) the ensemble yield of *Miscanthus* and Switchgrass with the yield from LPJml minus that from RF.

**A brief description of the algorithms in crop growth sub module of H08**

To make it clear for the function of the parameters we calibrated, here we briefly describe the algorithms in the crop growth sub module of H08. The crop module of H08 accumulates daily heat units ($Huna(t)$), which are expressed as the daily mean air temperature ($T_a$) greater than the plant's specific base temperature ($Tb$; given as a crop specific parameter):

$$Huna(t) = T_a - Tb \tag{1}$$

Then the heat unit index ($Ihun$) is calculated as the ratio of accumulated daily heat units $\sum Huna(t)$ and the potential heat unit ($Hun$):

$$Ihun = \frac{\sum Huna(t)}{Hun} \tag{2}$$

When the accumulated daily heat units $\sum Huna(t)$ reach the potential heat unit ($Hun$) required for the maturity of the crop, the crop is mature and is harvested. During the growth period, the daily increase in biomass ($\Delta B$) is calculated using a simple photosynthesis model:

$$\Delta B = be * PAR * REGF \tag{3}$$

Where $be$ is radiation use efficiency, $PAR$ is photosynthetically active radiation, and $REGF$ is the crop regulating factor. $PAR$ is calculated using shortwave radiation ($Rs$) and leaf area index ($LAI$) as follow:

$$PAR = 0.02092 * Rs * [1 - \exp(-0.65 * LAI)] \tag{4}$$

LAI is calculated according to the growth stage indicated by $Ihun$, if $Ihun < \lfloor dpl1 \rfloor * 0.01$,

$$LAI = \frac{(dpl1 - \lfloor dpl1 \rfloor) * Ihun}{\lfloor dpl1 \rfloor * 0.01} * blai \tag{5}$$

if $\lfloor dpl1 \rfloor * 0.01 \le Ihun < \lfloor dpl2 \rfloor * 0.01$,

$$LAI = \left\{ (dpl1 - \lfloor dpl1 \rfloor) + \frac{[(dpl2 - \lfloor dpl2 \rfloor) - (dpl1 - \lfloor dpl1 \rfloor)] * (Ihun - \lfloor dpl1 \rfloor * 0.01)}{\lfloor dpl2 \rfloor * 0.01 - \lfloor dpl1 \rfloor * 0.01} \right\} * blai \tag{6}$$

if $\lfloor dpl2 \rfloor * 0.01 \le Ihun < dlai$,

$$LAI = \left\{ (dpl2 - \lfloor dpl2 \rfloor) + \frac{[1 - (dpl2 - \lfloor dpl2 \rfloor)] * (Ihun - \lfloor dpl2 \rfloor * 0.01)}{dlai - \lfloor dpl2 \rfloor * 0.01} \right\} * blai \tag{7}$$

if $dlai < Ihun,$

85 $LAI = 16 * blai\,(1 - Ihun)^2$ (8)

$REGF$ is calculated as:

$REGF = \min(Ts, Ws, Ns, Ps)$ (9)

Where $Ts, Ws, Ns, Ps$ is respectively the stress factors for temperature, water, nitrogen, and phosphorous. Temperature stress

($Ts$) is calculated as an asymmetrical function according to the relationship between air temperature ($Ta$) and optimal

90 temperature ($To$). When air temperature is below (or equal) optimal temperature ($To$), $Ts$ is calculated as:

$Ts = exp\{ln(0.9) * [\frac{Ctsl(To-Ta)}{Ta}]^2\}$ (10)

Where $Ctsl$ is the temperature stress parameter for temperature below to, and is calculated as:

$Ctsl = \frac{To+Tb}{To-Tb}$ (11)

When air temperature is above optimal temperature, $Ts$ is calculated as:

95 $Ts = exp\{ln(0.9) * [\frac{(To-Ta)}{Ctsh}]^2\}$ (12)

Where $Ctsh$ is the temperature stress parameter for temperature below to, and is calculated as:

$Ctsh = 2 * To - Ta - Tb$ (13)

Water stress ($Ws$) is calculated as the ratio of actual evapotranspiration ($Ea$) to potential evapotranspiration ($Ep$) as:

$Ws = \frac{Ea}{Ep}$ (14)

100 As for nitrogen and phosphorous stress, currently we take it as neglectable since the bioenergy crop yield simulated by H08 is

with no constrains of nutrient.

The crop yield ($Yld$) is finally estimated by the aboveground biomass ($Bag$) with crop specific harvest index ($Harvest$) at

the harvesting date as:

$Bag = [1 - (0.4 - 0.2 * Ihun)]\sum \Delta B$ (15)

105 $$Yld = Harvest * \frac{WSF}{WSF + \exp(6.117 - 0.086*WSF)} * Bag \qquad (16)$$

Where *WSF* is a ratio of *SWU* (the accumulated actual plant transpiration in the second half of the growing season), and *SWP* (the accumulated potential evapotranspiration accumulated actual plant transpiration):

$$WSF = \frac{SWU}{SWP} * 100 \qquad (17)$$

---

## Author Response (AR2)

Dear Dr. Hisashi Sato,

Thank you very much for taking time to handle our paper (gmd-2020-179). We have the pleasure of enclosing a revised version of the manuscript titled "Simulating second-generation herbaceous bioenergy crop yield using the global hydrological model H08 (v.bio1)" and a detailed response to the Reviewers' comments below. We hope the manuscript is now suitable for publication in *Geoscientific Model Development*.

In the responses below, we have addressed all the comments from the anonymous Reviewer 2. The comments from the Reviewer are noted as "R" (e.g., R2) while each comment is noted as "C" (e.g., C1) to better index all comments. In the manuscript (with track changes), the updates are colored in red and the deletions are strikethrough in blue. Please do not hesitate to contact us if any further information is needed.

Sincerely yours,
Zhipin Ai (on behalf of co-authors)

**Response to Reviewer 2**

| No. | Comment | Response |
|---|---|---|
| R2C1 | The authors have addressed my previous comments. I only have some minor comments to help further improve the manuscript. | Dear Reviewer, thank you very much for taking time to have a second review. We are pleased to see your agreement on our revision. We have now further improved all the figures as you suggested here. Please find our responses to each comment below. |
| R2C2 | Figure 1: Is the "Land surface model" in Figure 1a "H08"? If so, it will be better to show H08 directly. | H08 contains all six submodules shown in Fig. 1a. Land surface module is only one of the submodules. Therefore, it is better to not show H08 there. |
| R2C3 | Figure 2: Try to add the names for the climate zones (e.g., tropical, dry, temperate) directly under the color bar, rather than showing numbers (e.g., 0.5, 1.5, 2.5) under the color bar and explain the names in the caption. | We have taken your good suggestion and added names for each climate zone (tropical, dry, temperate, continental, and polar) directly under the color bar in Fig. 2. |
| R2C4 | Figures 3 and 4: It will be great if the authors can keep the magnitude in the y-axis consistent for Miscanthus and switchgrass, respectively, to give readers a better visualization. For example, consistently using 60 as maximum for (a) and (c) and using 40 or 35 for (b) and (d). Also, using "(a) Miscanthus" as main titles. Same for the other figures. | We have taken your good suggestion and consistently set the maximum yield at 60 Mg ha$^{-1}$ yr$^{-1}$ for *Miscanthus* (Fig. 3a, Fig. 4a and 4c) and 35 Mg ha$^{-1}$ yr$^{-1}$ for switchgrass (Fig. 3b, Fig. 4b and 4d). We also modified the subtitles as you suggested in Figs. 3 and 4. We checked the other figures and revised the color bar in Fig. 9a and 9b to keep it consistent with Fig. 9c and 9d. |
| R2C5 | Figure 7: Using a consistent maximum y-axis value for all the bar plots and scatterplots, for better visualization. | We have taken your suggestion and consistently set the maximum y-axis to 35 Mg ha$^{-1}$ yr$^{-1}$ for all the bar plots and scatterplots in Fig. 7. |
| R2C6 | Figures 8 and 9: the subtitles are a little difficult to tell right now. How about put them in top middle? | We have taken your suggestion and moved the subtitle in the top middle in Figs. 8 and 9. |
| R2C7 | Substantively, the paper is in good shape. I appreciate the efforts the authors put into revising the manuscript. | We'd like to give our sincerely appreciate to the Reviewer for taking time to read our manuscript carefully and providing us many insightful comments and suggestions, which helped us a lot to improve the quality of our paper. |

[revised manuscript text omitted]